# Decision-Focused Learning with Directional Gradients

**Vishal Gupta**
USC Marshall School of Business
Los Angeles, CA 90029
guptavis@usc.edu

**Michael Huang**
CUNY Baruch Zicklin School of Business
New York, NY 10010
michael.huang@baruch.cuny.edu

## Abstract

We propose a novel family of decision-aware surrogate losses, called Perturbation Gradient (PG) losses, for the predict-then-optimize framework. The key idea is to connect the expected downstream decision loss with the directional derivative of a particular plug-in objective, and then approximate this derivative using zeroth order gradient techniques. Unlike the original decision loss which is typically piecewise constant and discontinuous, our new PG losses is a Lipschitz continuous, difference of concave functions that can be optimized using off-the-shelf gradient-based methods. Most importantly, unlike existing surrogate losses, the approximation error of our PG losses vanishes as the number of samples grows. Hence, optimizing our surrogate loss yields a best-in-class policy asymptotically, even in misspecified settings. This is the first such result in misspecified settings, and we provide numerical evidence confirming our PG losses substantively outperform existing proposals when the underlying model is misspecified.

## 1 Introduction

We study the contextual optimization problem

$$\pi^*(X) \in \arg\min_{z \in \mathcal{Z}} f^*(X)^\top z, \;\; f^*(X) \equiv \mathbb{E}\left[Y \mid X\right], \tag{1}$$

where $(X, Y) \in \mathcal{X} \times \mathcal{Y}$ are random variables, and $\mathcal{Z} \subseteq \mathbb{R}^d$ is a known, potentially non-convex feasible region. We work in a data-driven setting in which $f^*$ is unknown, but we observe i.i.d. draws $\{(X_i, Y_i) : i = 1, \ldots, n\}$ of $(X, Y)$. Problem (1) models applications in which we observe a potentially informative context $X$ before selecting the decision $\pi(X)$ such as vehicle routing, portfolio allocation, and inventory management [7, 5, 32]. Problem (1) has also been used as an "optimization layer" in neural network architectures to model combinatorial decisions [26]. Through a suitable transformation, it can also represent some, but not all, nonlinear problems like personalized pricing (see Appendix A).

The predict-then-optimize framework focuses on *plug-in policies* for Problem (1). Given a function $f : \mathcal{X} \mapsto \mathcal{Y}$, the corresponding plug-in policy is

$$\hat{\pi}(f(X)) \in \arg\min_{z \in \mathcal{Z}} f(X)^\top z, \tag{2}$$

with ties broken by some pre-specified tie-breaking rule. Plug-in policies are attractive because they separate the prediction procedure ($f$) from the optimization procedure (Problem (2)). This decoupling is especially useful when i) decisions $z$ must satisfy hard constraints (enforced by $\mathcal{Z}$), or ii) one has a specialized algorithm for solving instances of Problem (2) (e.g., a custom vehicle-routing solver).

Given the form of $\pi^*$, a natural approach might be to learn an estimate $\hat{f}$ of $f^*$ from the data, e.g., by minimizing the mean-squared error, and then compute $\hat{\pi}(\hat{f}(X))$. Such procedures are called *decision-blind* since they do not leverage Problem (1) when learning $\hat{f}$.

38th Conference on Neural Information Processing Systems (NeurIPS 2024).

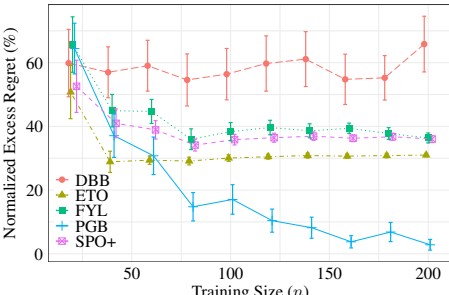

Figure 1: (Convergence under Misspecification). Excess regret normalized by optimal policy's performance under the misspecified setting of Section 4.1 ($\alpha = 1$, $m = 0$). PGB is our proposed loss. ETO is a decision-blind approach that minimizes MSE. Other benchmarks include: DBB [26], FYL [1], and SPO+ [7]. Under misspecification, only the PG losses have vanishing excess regret. Error bars are 95% confidence intervals on the mean over 100 trials.

The seminal paper Elmachtoub and Grigas [7] argues *decision-aware* techniques can be superior to decision-blind ones. Given a hypothesis class $\mathcal{F} \subseteq \mathcal{Y}^{\mathcal{X}}$, they propose solving $\min_{f \in \mathcal{F}} \text{Regret}(f)$ where $\text{Regret}(f) \equiv \mathbb{E}\left[Y^\top \hat{\pi}(f(X))\right] - \mathbb{E}\left[Y^\top \hat{\pi}(f^*(X))\right]$. This is equivalent to solving

$$\min_{f \in \mathcal{F}} \mathbb{E}\left[\ell(f(X), Y)\right] \quad \text{where } \ell(t, y) \equiv y^\top \hat{\pi}(t). \tag{3}$$

Growing empirical evidence supports the strength of decision-aware approaches [31, 27].

A challenge is that when $\mathcal{Z}$ is polyhedral or combinatorial, $t \mapsto \ell(t, y)$ is a piecewise constant, discontinuous map. Its gradient is either zero or undefined at all points. Hence, one cannot easily apply a first-order method like stochastic gradient descent (SGD) to optimize Problem (3).

In this paper we propose a new family of surrogate losses to approximate $\ell(t, y)$ by connecting $\ell(t, y)$ to the directional derivative of a particular plug-in function and using zeroth order gradients to approximate this derivative. We call this family *perturbation gradient (PG) losses*. PG losses are Lipschitz continuous, general purpose, and only require a black-box oracle which solves Problem (2). Most importantly, their gradients are "informative" (c.f. Lemma 2.2); after replacing $\ell$ with a PG loss, one can apply a first order method to Problem (3) or its empirical counterpart "out-of-the-box."

Previous authors have also proposed surrogates which satisfy some of these properties (see Section 1.2). What distinguishes our work is that under mild assumptions on the distribution of $(X, Y)$, the error of our surrogate in approximating $\ell(t, y)$ vanishes as $n \to \infty$ with a rate that depends on the complexity of $\mathcal{F}$. More precisely, we prove that, for general $\mathcal{Z}$, optimizing the empirical PGB loss (a particular member of the PG family) induces an excess regret over the best-in-class policy of at most $\tilde{O}_p(\sqrt{\mathfrak{R}^n} + n^{-1/2})$ where $\mathfrak{R}^n$ is the multivariate Rademacher complexity of $\mathcal{F}$ (Theorem 3.8). For linear hypotheses with $\dim(X) = p$, this bound reduces to $\tilde{O}_p((dp/n)^{1/4})$. When $\mathcal{Z}$ is polyhedral, optimizing empirical PGB loss induces an excess regret of at most $\tilde{O}_p(n^{-1/2}\sqrt{\nu \log |\mathcal{Z}_\angle|})$, where $\nu$ is VC linear subgraph dimension of $\mathcal{F}$ and $\mathcal{Z}_\angle$ are the extreme points of $\mathcal{Z}$ (Theorem 3.8). Both bounds vanish as $n \to \infty$, implying that optimizing our PGB loss yields a best-in-class policy asymptotically.

Critically – and this is the most distinctive feature of our work – our results hold even when $f^* \notin \mathcal{F}$ (misspecified setting). To our knowledge, these are the first result of their kind for a differentiable surrogate. Existing results on the predict-then-optimize framework [19, 14, 8]) require $f^* \in \mathcal{F}$ (the well-specified setting) and somewhat restrictive assumptions on the noise $Y - f^*(X)$ (see Section 1.2). These requirements are not simply a weakness in prior analysis. As seen in Fig. 1, existing methods can have very poor performance under misspecification. The key issue is that the justification for many of these losses tacitly relies on the fact that an optimal $f$ should be such that $f(X) \approx Y$ almost surely, but under misspecification, this is generally impossible. Hence, they do not well-approximate the decision loss $\ell$. See Fig. 2.

This poor performance is especially unfortunate, because misspecified settings are *precisely* those where decision-aware learning offers the most benefit over decision-blind approaches [8, 3]. This is for at least three reasons: First, because the solution mapping $\hat{\pi}(\cdot)$ is piecewise constant, there may exist $f \neq f^*$ such that $\hat{\pi}(f(X)) = \hat{\pi}(f^*(X))$ almost surely [7, 34]. (Indeed, this appears to occur in Fig. 1.) Hence, one might still achieve (near) zero regret by learning over a low-complexity $\mathcal{F}$ in a decision-aware fashion, and, typically low-complexity hypothesis classes $\mathcal{F}$ are preferred for tractability, interpretability, and strong generalization properties. Second, when every $f$ must induce some error, decision-aware learning seeks an $f(\cdot)$ such that $\hat{\pi}(f(X))$ disagrees with $\hat{\pi}(f^*(X))$ on regions of the covariate space $\mathcal{X}$ that are not too costly in the decision-problem, while decision-blind methods typically seek an $f$ such that $f(X)$ disagrees with $f^*(X)$ on less probable regions of $\mathcal{X}$

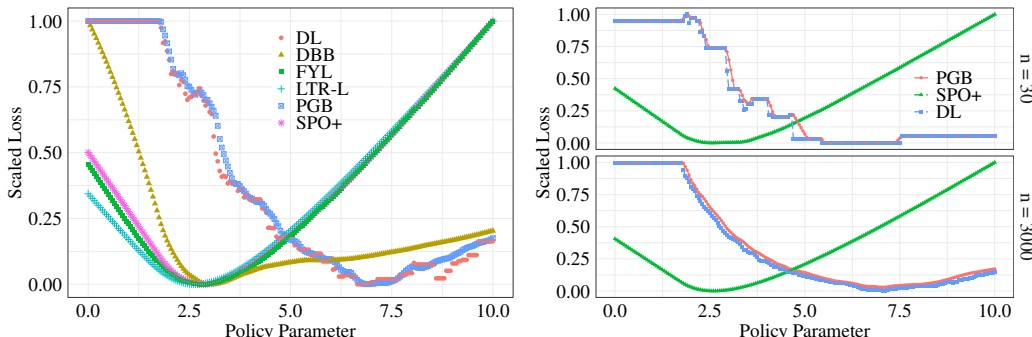

Figure 2: (Comparing Surrogates under Misspecification). See Section 4.1 for setup ($\alpha = 1$, $m = 0$). Benchmarks are decision-loss (DL) $\ell$, our PGB and PGC losses, Fenchel-Young Loss (FYL) [1], SPO+ [7], and the learning-to-rank list loss ([21]. Left-panel: ($n = 200$) Only our PG losses closely track the DL. Right Panels: As $n$ increases, the DL and PG losses both become smoother.

[3]. Finally, [8, 14] suggest traditional decision-blind learning strictly dominates decision-aware techniques in a well-specified setting (see Section 1.2), i.e, decision-aware learning loses most of its advantages if $f^* \in \mathcal{F}$.

To be fair, the improved approximation quality of our PG losses comes at the cost of computational complexity. Many existing surrogates are convex if $\mathcal{F}$ is a linear class. We can optimize these surrogates over $\mathcal{F}$ in polynomial time. On the other hand, Elmachtoub and Grigas [7] shows that solving the empirical counterpart of Problem (3) is NP-Hard (reduction from $0/1$ classification). Thus, these aforementioned surrogates cannot be expected to reliably find best-in-class policies without additional assumptions on the data distribution unless P = NP.

By contrast, our proposed PG losses are non-convex, expressible as the difference of concave functions. Optimizing such functions is well-studied [25, 30, 18], but is, in the worst-case, NP-Hard. This is to be expected; if we seek a method that finds a best-in-class policy, it must contend with this hardness. Importantly, some NP-Hard problems admit algorithms that find high-quality solutions efficiently for most real-world instances. We argue our loss (combined with simple gradient descent type methods) yields such a problem. Previous authors [26, 34] have also proposed non-convex surrogates and shown that first-order methods still recover high-quality solutions.

Finally, we offer that convexity is often moot in applications. When using a nonlinear hypothesis class (e.g., a neural network with more than 1 layer), even convex surrogates induce non-convex loss functions. Optimizing these losses is theoretically no easier than optimizing our surrogate.

In summary, our PG losses represent a practically implementable and (the first) theoretically justified approach to decision-aware learning in misspecified setting.

### 1.1 Contributions

- We propose a new family of surrogate losses called Perturbation Gradient (PG) losses for the predict-then-optimize approach to Problem (1). Our surrogates are Lipschitz continuous and can be expressed as the difference of concave functions.
- We show that the gradient of a PG loss evaluated at a sample point is an unbiased estimate of the gradient of the expected loss (Lemma 2.2). Thus, unlike the decision loss $\ell$, we can apply first-order methods to minimize our expected surrogate or its ERM counterpart.
- We bound the uniform approximation error of our surrogates with respect to decision loss by a term vanishing in $n$ (Theorems 3.4 and 3.7). Thus, with more data, our loss becomes more accurate.
- We prove that the empirical minimizer of our PGB loss yields a best-in-class policy asymptotically, even if the underlying hypothesis class is misspecified (Theorem 3.8). To our knowledge, ours is the *first* surrogate for the predict-then-optimize framework with such a performance guarantee.
- We provide numerical evidence showing that minimizing our surrogate loss performs comparably to other surrogates when the hypothesis class is well-specified, and substantively outperforms them when the hypothesis class is misspecified.

## 1.2 Related Work

Elmachtoub and Grigas [7] first proposed a convex, differentiable surrogate loss for Problem (3) called the SPO+ loss leveraging a duality argument. Subsequent researchers have proposed other approaches to surrogate creation including replacing the plug-in policy Problem (2) with a regularized counterpart [32], creating a response-surface [28, 10], training a neural network to approximate $\ell(t, y)$ non-parametrically [34], linearizing $l(t, y)$ [26], and combining randomized-smoothing with conjugate duality [1]. A recent computational study [31] compares many of these approaches and found that SPO+ and the Fenchel-Young loss of [1] performed best or near-best on all benchmarks.

Despite the empirical strengths of decision-aware methods, their theoretical justification is less clear. Few methods establish regret bounds. Wilder, Dilkina, and Tambe [32] and Berthet et al. [1] prove that gradients of particular surrogates can be evaluated easily, but do not prove a regret guarantee for the minimizer of those surrogates. On the other hand, El Balghiti et al. [6] and Hu, Kallus, and Mao [14] prove generalization guarantees relating $\mathbb{E}\left[\ell(f(X), Y)\right]$ to its empirical counterpart; hence, if one finds an $f \in \mathcal{F}$ with small empirical loss, $\mathbb{E}\left[l(f(X), Y)\right]$ will also be small. But minimizing the empirical counterpart to Problem (3) is computationally challenging. Jeong et al. [16] proposes a symbolic reduction scheme for this task. However, the method only applies to linear $f$ and does not scale to large $n$. Most importantly, it is not amenable to first-order methods, so cannot be easily integrated into neural architectures.

The strongest known regret bounds are for the SPO+ loss in the well-specified setting ($f^* \in \mathcal{F}$). When the conditional distribution of $Y|X$ is centrally symmetric around its mean, Elmachtoub and Grigas [7] establish a Fisher-consistency result. Liu and Grigas [19] strengthen this result, establishing (under similar assumptions) that if the multivariate Rademacher complexity of $\mathcal{F}$ is $O(n^{-1/2})$, then the empirical minimizer of the SPO+ loss has regret at most $O(n^{-1/4})$.

That said, such results are perhaps unsatisfying because decision-blind methods typically dominate decision-aware methods in well-specified settings. Hu, Kallus, and Mao [14] show that when $f^* \in \mathcal{F}$, the regret of a decision-blind approach that minimizes MSE converges to zero faster than the empirical minimizer of Problem (3). Said differently, decision-aware methods likely offer the most benefit in misspecified settings. Hence, these settings are arguably the most interesting.

Most closely related to our work are perturbation-based approaches for estimating out-of-sample performance. These works each use Danskin's theorem to "debias" a naive estimate of out-of-sample performance. Ito, Yabe, and Fujimaki [15] and Guo, Jordan, and Zhou [11] each establish asymptotic convergence of their estimators (without an explicit rate): Ito, Yabe, and Fujimaki [15] treats a non-contextual setting and focuses on the ERM estimator. Guo, Jordan, and Zhou [11] treats a causal inference setting. By contrast, Gupta, Huang, and Rusmevichientong [12, 13] establish a finite-sample regret guarantee, but in a small-data, large-scale data regime with nearly-Gaussian corruptions. In this paper, we focus on the traditional large-sample regime ($n \to \infty$) with contexts. Moreover, instead of "debiasing," we perturbations to approximate a directional derivative which exactly represents our out-of-sample loss.

## 1.3 Notation and Preliminaries

We write $a \lesssim b$ to mean that there exists a universal constant $C$ such that $a \leq Cb$. We denote the $\ell_2$ norm by $\|\cdot\|$. To simplify the presentation, we also make the following boundedness assumption:

**Assumption 1.1** (Boundedness). There exists $B > 0$ such that $\max_{z \in \mathcal{Z}} \|z\| \leq B$, and $\|Y\| \leq 1$, almost surely.

## 2 A New Family of Surrogate Losses

Define the plug-in policy objective:

$$V(t) = \min_{z \in \mathcal{Z}} t^\top z = t^\top \hat{\pi}(t).$$

Evaluating $V(t)$ only requires a black-box oracle for Problem (2). Since it is minimum of linear functions, $V(t)$ is concave.

Our first key observation is that by Danskin's Theorem [2, Prop B.22],

$$\frac{\partial}{\partial \lambda} V(t + \lambda y) \mid_{\lambda=0} = y^\top \hat{\pi}(t) = \ell(t, y), \tag{4}$$

where the left side is a derivative if $\hat{\pi}(t)$ is unique and a subgradient otherwise. We can form a family of PG surrogates by considering different zeroth order approximations to the derivative on the left (see [20, 24] for more on zeroth order gradients). We focus on two specific zeroth order gradients:

- Backward Differencing (PGB): $\hat{\ell}_h^b(t, y) \equiv \frac{1}{h} \left( V(t) - V(t - hy) \right)$
- Central Differencing (PGC): $\hat{\ell}_h^c(t, y) \equiv \frac{1}{2h} \left( V(t + hy) - V(t - hy) \right),$

for some user-defined $h > 0$. Intuitively, as $h \to 0$, both $\hat{\ell}_h^b(t, y)$ and $\hat{\ell}_h^c(t, y)$ should better approximate $\ell(t, y)$. (We formalize the tradeoff in $h$ below.)

For intuition on the shape of PG losses, consider the special case where $\mathcal{Z} = [-1, 1]$, and $Y \in \{-1, 1\}$. Then, $\ell(t, y) = -\text{sgn}(ty)$, a step function. The PGB and PGC losses are both ramp losses in this case, where the width of the ramp is controlled by $h$.

Other zeroth order gradient schemes are possible. For example, forward differencing yields the surrogate from [26], motivated from a different perspective. This alternate perspective sheds light on empirical performance. Indeed, our theoretical analysis suggests $h$ should be small, tending to zero, while [26] advocates for large $h$. Our analysis also shows forward differencing suffers optimistic bias because it overestimates the derivative of a concave function. These features might explain the poor performance of [26] in [31] benchmarks. We explore some of these issues in Appendix B, but fully characterizing how the choice of zeroth order gradient affects surrogate quality is an open problem.

## 2.1 Properties of PG Losses

Using the structure of Problem (1), we prove some key properties of our surrogates.

**Lemma 2.1** (Basic Properties). *Suppose Assumption 1.1 holds. For any $t, t' \in \mathbb{R}^d$ and $y \in \mathcal{Y}$, the PG losses are*

a) *Lipschitz, i.e.,* $\left| \hat{\ell}^b(t, y) - \hat{\ell}^b(t', y) \right| \leq \frac{2B}{h} \|t - t'\|$, *and* $\left| \hat{\ell}^c(t, y) - \hat{\ell}^b(t', y) \right| \leq \frac{B}{h} \|t - t'\|$.

b) *Bounded, i.e.,* $\left| \hat{\ell}^b(t, y) \right| \leq B$, *and* $\left| \hat{\ell}^c(t, y) \right| \leq B$.

c) *Differentiable* [1], *i.e.,* $\nabla_t \hat{\ell}^b(t, y) = \frac{1}{h}(\hat{\pi}(t) - \hat{\pi}(t - hy))$, *and* $\nabla_t \hat{\ell}^c(t, y) = \frac{1}{2h}(\hat{\pi}(t + hy) - \hat{\pi}(t - hy))$.

*Finally, the backward difference upperbounds the true loss, i.e., $\ell(t, y) \leq \hat{\ell}^b(t, y)$.*

A primary advantage of our PG losses over the original loss $\ell$ is that gradients are "informative." More precisely, because $\ell$ is discontinuous, $\nabla_t \mathbb{E}\left[\ell(t, Y)\right] \neq \mathbb{E}\left[\nabla_t \ell(t, Y)\right]$, and $\nabla_t \ell(t, Y_j)$ is not an unbiased estimate of $\nabla_t \mathbb{E}\left[\ell(t, Y)\right]$. Our surrogates do not have this problem.

**Lemma 2.2** (Informative Gradients). *Suppose Assumption 1.1 holds. For all $t$ and $Y$, $\nabla_t \mathbb{E}[\hat{\ell}_h^b(t, Y)] = \mathbb{E}[\nabla_t \hat{\ell}_h^b(t, Y)]$. Thus, $\nabla_t \hat{\ell}_h^b(t, Y_j)$ is an unbiased estimate of $\nabla_t \mathbb{E}\left[\hat{\ell}_h^b(t, Y)\right]$. The same statements also hold $\hat{\ell}_h^c$.*

Lemma 2.2 ensures that we can apply first order methods out-of-the-box to optimize our PG losses.

## 3 Performance Guarantees

For brevity, we focus on the backward PG loss. Analogous results hold for the central PG loss.

**Key Intuition.** The key challenge is bounding the error between our PGB loss $\hat{\ell}_h^b$ and the decision loss $\ell$. For intuition, consider the expected error at a fixed $f \in \mathcal{F}$, i.e., $\mathbb{E}\left[\hat{\ell}_h^b(T, Y) - \ell(T, Y)\right]$, where

---

[1]These expressions are gradients when $\hat{\pi}(t)$ and $\hat{\pi}(t \pm y)$ are unique optimizers, and elements of the Clarke subdifferential otherwise.

$T = f(X)$. Define the auxiliary function $H(\lambda) = \mathbb{E}\left[V(T + \lambda Y)\right]$. When $\hat{\pi}(T + \lambda Y)$ is unique, Lemma E.1 justifies switching the derivative and expectation yielding

$$H'(\lambda) = \mathbb{E}\left[\frac{d}{d\lambda}V(T + \lambda Y)\right] = \mathbb{E}\left[Y^{\top}\hat{\pi}(T + \lambda Y)\right],$$

where the last equality is Danskin's theorem [2, Prop B.22]. Thus, $\mathbb{E}\left[\hat{\ell}^{b}(T, Y) - \ell(T, Y)\right] = \frac{1}{h}(H(0) - H(-h)) - H'(0)$, i.e., the expected approximation error equals the error in estimating the derivative of $H$.

If $H$ is not sufficiently well-behaved, this error may not be small. Lemma E.2 proves that if $H$ is $\beta$-smooth, i.e., $H'(\lambda)$ is $\beta$-Lipschitz, then this error is at most $\beta h$. Since $H$ entails expectation, we intuit that it should be smooth if $(T, Y)$ has a "nice" density, similar to the intuition behind randomized smoothing.

To quantify what "nice" might mean, write

$$\left|H'(\lambda) - H'(\bar{\lambda})\right| = \left|\mathbb{E}\left[Y^{\top}\hat{\pi}(T + \lambda Y)\right] - \mathbb{E}\left[Y^{\top}\hat{\pi}(T + \bar{\lambda}Y)\right]\right|.$$

Since $(t, y) \mapsto Y^{\top}\hat{\pi}(T + \lambda Y)$ is $B$-bounded by Lemma 2.1, the last difference is at most $B \cdot TV((Y, T + \lambda Y), (Y, T + \bar{\lambda}))$, where $TV(\cdot, \cdot)$ is the total variation distance between the two random vectors. Hence, a "nice" density is any density such that distributions of $(Y, T + \lambda Y)$ and $(Y, T + \bar{\lambda}Y)$ are close whenever $\lambda$ and $\bar{\lambda}$ are close. We expect this generally occurs whenever $(T, Y)$ admit Lipschitz continuous densities, but can be shown to fail if, e.g., $T$ is concentrated at a single point.

We make the above intuition formal in the next section.

## 3.1 Expected Approximation Error

We make the following assumption:

**Assumption 3.1** (Lipschitz Log Conditional Density). *Let $g(\cdot\,; f, Y)$ be the conditional density of $f(X) \mid Y$. We assume that there exists a constant $L > 0$ such that $\log g(\cdot\,; f, Y)$ is $L$-Lipschitz for all $f \in \mathcal{F}$ and all $Y$ almost surely.*

As discussed above, Assumption 3.1 is sufficient to ensure the requisite TV distance is small, but not necessary. We prefer Assumption 3.1 as it facilitates a short proof. Under this assumption, we have:

**Lemma 3.2** (Expected Approximation Error). *Suppose Assumptions 1.1 and 3.1 hold and $h < \frac{1}{L}$. Then, for any $f \in \mathcal{F}$, $0 \leq \mathbb{E}[\hat{\ell}_{h}^{b}(f(X), Y) - \ell(f(X), Y)] \leq (e - 1)B \cdot L \cdot h$.*

## 3.2 Uniform Error Bounds

Combining Lemma 3.2 and Hoeffding's inequality, yields a pointwise bound:

**Corollary 3.3** (Pointwise Approximation Error). *Fix some $f \in \mathcal{F}$. Suppose Assumptions 1.1 and 3.1 hold and $h < \frac{1}{L}$. Then, for any $0 < \delta < \frac{1}{2}$, with probability at least $1 - \delta$,*

$$\left|\frac{1}{n}\sum_{j=1}^{n}\hat{\ell}_{h}^{b}(f(X_j), Y_j) - \mathbb{E}\left[\ell(f(X), Y)\right]\right| \lesssim BLh + B\sqrt{\log(1/\delta)/n}.$$

As seen in Lemma 2.1, the Lipschitz constant of $\hat{\ell}_{h}^{b}$ scales like $1/h$. Hence, unlike other learning methods, $h$ does *not* control a bias-variance tradeoff; rather $h$ controls a bias-computational complexity tradeoff. Practically, we suggest taking $h$ as large as the next largest term in the bound, i.e. $h = O(n^{-1/2})$ above, to maximize the smoothness without compromising the rate.

Corollary 3.3 captures the key ideas of our approach, but is insufficient to establish a regret guarantee; we need a uniform error bound. To that end, we prove two results:

Our first generalization bound applies to any choice of $\mathcal{Z}$. We leverage the Lipschitzness of $\hat{\ell}_{h}^{b}$ (Lemma 2.1a) to apply a vector contraction inequality from Maurer [22] and bound the Rademacher complexity of our sample surrogate loss. A similar strategy is used in [19].

More specifically, define the multivariate Rademacher complexity

$$\mathfrak{R}^{n}\left(\mathcal{F}\right) = \mathbb{E}\left[\hat{\mathfrak{R}}^{n}\left(\mathcal{F}\right)\right] = \mathbb{E}\left[\sup_{f \in \mathcal{F}}\frac{1}{n}\sum_{i=1}^{n}\boldsymbol{\sigma}_{i}^{\top}f(X_i)\right], \tag{5}$$

where $\boldsymbol{\sigma}_i = (\sigma_{i1}, \ldots, \sigma_{id})$ and $\sigma_{ij}$ are i.i.d. Rademacher random variables. Then, we have

**Theorem 3.4** (Uniform Error Bound for General $\mathcal{Z}$). *Suppose Assumptions 1.1 and 3.1 hold. For any $0 < \delta < \frac{1}{2}$ and $0 < h < \frac{1}{L}$, with probability at least $1 - \delta$*

$$\sup_{f \in \mathcal{F}} \left| \frac{1}{n} \sum_{i=1}^{n} \hat{\ell}_h^b \left( f(X_i), Y_i \right) - \mathbb{E}\left[ \ell \left( f(X), Y \right) \right] \right| \lesssim BLh + \frac{B^2}{h} \mathfrak{R}^n(\mathcal{F}) + B\sqrt{\log(1/\delta)/n}.$$

If $\dim(X) = p$ and $\mathcal{F}$ is a linear class, $\mathfrak{R}^n(\mathcal{F}) = \tilde{O}(\sqrt{dp/n})$ [6]. Choosing $h = O((dp/n)^{1/4})$ yields an error of size $\tilde{O}_p((dp/n)^{1/4})$. This is same rate as Liu and Grigas [19], but also holds in the misspecified setting where $f^* \notin \mathcal{F}$.

Theorem 3.4 applies to general $\mathcal{Z}$, but may be loose. We next present a stronger result when $\mathcal{Z}$ is polyhedral by leveraging results from Hu, Kallus, and Mao [14] based on VC dimension:

**Definition 3.5** (VC-Linear-Subgraph Dimension). The VC-linear-subgraph dimension of a class of functions $\mathcal{F} \subseteq \mathcal{Y}^{\mathcal{X}}$, is the VC dimension of the sets $\mathcal{F}^\circ = \left\{ \left\{ (x, \beta, t) : \beta^\top f(x) \leq t \right\} : f \in \mathcal{F} \right\}$ in $\mathcal{X} \times \mathbb{R}^{d+1}$, that is, the largest integer $\nu$ for which there exist $x_1, \ldots, x_\nu \in \mathcal{X}$, $\beta_1, \ldots, \beta_\nu \in \mathbb{R}^d$, $t_1 \in \mathbb{R}, \ldots, t_\nu \in \mathbb{R}$ such that $\left| \left\{ \left( \mathbb{I} \left\{ \beta_j^\top f(x_j) \leq t_j \right\} : j = 1, \ldots, \nu \right) : f \in \mathcal{F} \right\} \right| = 2^\nu$.

We make the following assumption:

**Assumption 3.6** (Bounded VC Dimension). The VC-linear-subgraph dimension of the class $\bar{\mathcal{F}} = \left\{ \bar{f} : \bar{f}(x, y) = f(x) + hy, \text{ for } f \in \mathcal{F}, h \in \mathbb{R} \right\}$ is at most $\nu$.

We obtain the following bound for polyhedral $\mathcal{Z}$, where $\mathcal{Z}_\angle$ is the set of extreme points of $\mathcal{Z}$.

**Theorem 3.7** (Uniform Error Bound for Polyhedral $\mathcal{Z}$). *Suppose Assumptions 1.1, 3.1 and 3.6 hold. For any $0 < \delta < \frac{1}{2}$ and $0 < h < \frac{1}{L}$, with probability at least $1 - \delta$,*

$$\sup_{f \in \mathcal{F}} \left| \frac{1}{n} \sum_{i=1}^{n} \hat{\ell}_h^b \left( f(X_i), Y_i \right) - \mathbb{E}\left[ \ell \left( f(X_i), Y_i \right) \right] \right| \lesssim BLh + B\sqrt{\frac{\nu \log(|\mathcal{Z}_\angle| + 1) \log(1/\delta)}{n}}.$$

Choosing $h = O(n^{-1/2})$ yields a bound of size $O_p(n^{-1/2})$ which matches the generalization error of $\ell$ from [14, 6]. Thus, for polyhedral $\mathcal{Z}$, our surrogate converges no slower than the empirical loss, but is more computationally tractable.

### 3.3 Excess Regret Bounds

We next transform the uniform bounds of the previous section to bounds on excess regret. Define

$$\text{ERegret}(f) \equiv \mathbb{E}\left[ Y^\top \hat{\pi}(f(X)) \right] - \mathbb{E}\left[ Y^\top \hat{\pi}(f^{OR}(X)) \right], \quad \text{where} \quad f^{OR} \in \operatorname{argmin}_{f \in \mathcal{F}} \text{Regret}(f).$$

Excess regret measures regret relative to the best-in-class policy $f^{OR}$, not the full-information optimum $f^*$. For a fixed $h < \frac{1}{L}$, define the empirical minimizer of PGB loss $\hat{f}_h \in \operatorname{argmin}_{f \in \mathcal{F}} \frac{1}{n} \sum_{i=1}^{n} \hat{\ell}_h^b \left( f(X_i), Y_i \right)$. Then, we have the following:

**Theorem 3.8** (Excess Regret Bounds).

i) *Suppose the assumptions of Theorem 3.4 hold. Then,* $\text{ERegret}(\hat{f}_h) \lesssim \sqrt{B^3 L \mathfrak{R}^n(\mathcal{F})} + \frac{B}{\sqrt{n}}$.

ii) *Suppose the assumptions from Theorem 3.7 hold. Then,* $\text{ERegret}(\hat{f}_h) \lesssim B\sqrt{\frac{\nu \log(|\mathcal{Z}_\angle| + 1)}{n}}$.

For many hypothesis classes, the multivariate Rademacher complexity is vanishing in $n$. Hence, both bounds are vanishing in $n$ and $\hat{f}_h$ achieves best-in-class performance asymptotically.

## 4 Numerical Experiments

We compare learning a linear hypothesis class with our PG losses (PGB and PGC) to surrogates currently implemented in the PyEPO Python package [31]. Specifically, we benchmark against: SPO+ [7], DBB [26], FYL [1], and the family of LTR losses [21]. Additionally, we also benchmark against a "decision-blind" two stage policy that first minimizes the least-squares loss and then implements

the corresponding plug-in policy (ETO). We optimize each surrogate using ADAM via the PyEPO framework. All methods are trained for a total of 100 epochs, and we select the best model found in those 100 epochs based on validation set of size 200. For PG losses, we initialized at the SPO+ solution and choose $h$ from a small grid of values based on validation set performance. Future computational experiments might explore the effect of alternate initializations. We do not add additional regularization or smoothing to any of the surrogates. See Appendix C for other implementation details.

Our metric of interest is the normalized excess regret ($\mathbb{E}\left[Y^\top\left(\pi^*(X)-\hat{\pi}(X)\right)\right]/\mathbb{E}\left[Y^\top\pi^*(X)\right]$), where we have normalized by the optimum policy (c.f. Problem (1)) for interpretability.

## 4.1 Simple Misspecification Experiment

In our first experiment, we let $\mathcal{Z}\equiv\{0,1\}$. We let $X\sim\text{Unif}(0,2)$ and

$$f^*(x)=\begin{cases}-4x+2, & \text{for } x\in[0,0.55)\\ m(x-0.55)-0.2, & \text{for } x\in[0.55,2]\end{cases}$$

The function is piecewise linear with one piece that has a slope of $-4$ and another piece with a slope of $m\in[0,-4]$ (an elbow). The change point is at $x=0.55$ where the two functions meet at $-0.2$ (see Fig. 7 in Appendix D). Intuitively, $m$ controls the degree of misspecification; at $m=-4$, $f^*\in\mathcal{F}$ and the problem is well-specified. At $m=0$, the problem is maximally misspecified.

We generate synthetic data as $Y=f^*(X)+\epsilon_\alpha$. We define $\epsilon_\alpha=\sqrt{\alpha}\left(\zeta-0.5\right)+\sqrt{1-\alpha}\,\gamma$ where $\alpha\in[0,1]$, $\zeta$ is an exponential random variable with mean 0.5, and $\gamma\sim\mathcal{N}(0,0.25)$. By construction $\epsilon$ is mean-zero noise with variance 0.25. The value of $\alpha\neq0$, $\epsilon$ controls how asymmetric the noise is. Note, when $\alpha\neq0$, the theoretical performance guarantees on SPO+ from [19] do not apply.

**Results.** Figure 1 plots the relative regret for $m=0$ and $\alpha=1$, that is, the most misspecified setting with the most asymmetric noise $\epsilon$. Beyond highlighting the superior performance of the PG losses in misspecified settings, Fig. 1 also shows the choice of finite difference approximation (backward or central) also impacts performance. Intuitively, central differencing likely outperforms backward differencing because in standard, deterministic settings, central finite differencing has error $O(h^2)$ relative to the true derivative, while backward differencing has error $O(h)$ [17]. This intuition can be made formal in our setting by adapting Lemma 3.2, but we omit the details for brevity.

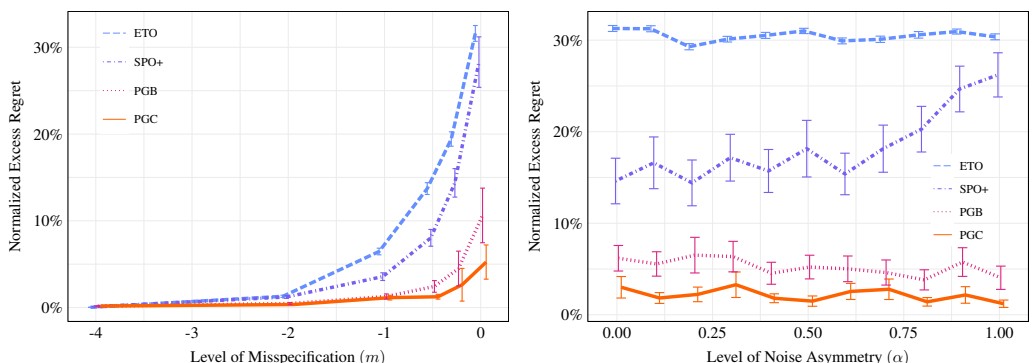

Figure 3: (SPO+ Comparisons) The left figure plots the excess regret normalized by the optimal policy's performance as we vary $m$ for $n=80$ and $\alpha=1$. The right figure plots the same value as we vary $\alpha$ for $n=200$. When $\alpha=0$ the noise is centrally symmetric and when $\alpha=1$ the noise is the most asymmetric. Error bars are $95\%$ confidence intervals on the mean over 100 trials.

The left panel of Fig. 3 in Appendix D studies the effect of increasing degrees of misspecification. We limit the benchmarks to ETO and SPO+ as other methods are qualitatively similar. We find that (as argued in the introduction), in well-specified settings ($m=-4$), the benefits of decision-aware learning may be small. All methods (including decision-blind ETO) achieve low regret, even for small $n$. In our experiments, even for $n=20$ the relative regret was less than $0.6\%$ across all methods.

On the other hand, as the degree of misspecification grows, decision-aware methods have an advantage. However, we see that SPO+ is nearly as susceptible as to misspecification as decision-blind approaches since the relative regret also increases rapidly. By contrast, the relative regret for our PG losses increases more slowly. We stress, this experiment fixes $n$. As $n \to \infty$, our theory shows the regret of the PG losses tends to best-in-class as in Fig. 1.

The right panel of Fig. 3 studies how the noise distribution impacts the relative regret since all prior known performance guarantees for SPO+ require strong assumptions on the noise [7, 19]. The plot suggests that requiring a symmetric noise is not simply a weakness in the analysis of SPO+, but fundamental to the method. As the noise becomes less symmetric, the performance of SPO+ degrades. Even when the assumption is satisfied ($\alpha = 0$), we see SPO+ is still significantly impacted by misspecification. By contrast, the PG losses perform similarly as the shape of the noise varies.

### 4.2 Shortest Path Experiments

**Random Arc Costs.** We first replicate the shortest path experiment from [7, 31] on a $5 \times 5$ grid graph. We let $X \sim \mathcal{N}(0, \boldsymbol{I}_5)$ and for each edge $j$, and take

$$f_j^*(x) = \frac{1}{3.5^6}\left[\left(\frac{1}{\sqrt{5}}(B^*x)_j + 3\right)^6 + 1\right]$$

where $B^* \in \{0, 1\}^{40 \times 5}$ has i.i.d. Bernoulli(0.5) entries (drawn once and fixed throughout). We consider two different data generation mechanisms: i) Multiplicative noise, i.e., $Y_j = f_j^*(X)(1 + \epsilon_j)$ where $\epsilon_j$ are i.i.d Unif$[-.3, .3]$. This choice closely mirrors the original experiment of [7]. ii) Additive Gaussian noise, i.e., $Y_j = f_j^*(X) + \varepsilon_j$ where $\varepsilon_j \sim \mathcal{N}(0, 0.3^2)$.

Figure 8 in the Appendix D compares the PG losses to the best two surrogates in our experiments, FYL [1] and SPO+ [7]. Here PGF represents a zeroth order gradient using forward differencing and is equivalent to the method of [26] but with a small $h$ as opposed to a large $h$. Despite the non-convexity, minimizing our PG losses with first order methods yields comparable performance to FYL and SPO+ (convex methods). In other words, they do not seem to get stuck in local minimima. For small $n$, we do seem some distinction, which is likely because our losses are less smooth (see the right figure of Fig. 2).

**Harder Example with Planted Arcs.** Because arc costs are completely at random in the previous example, there are likely many paths with near-optimal length. We next consider a harder instance where we hide a unique good path.

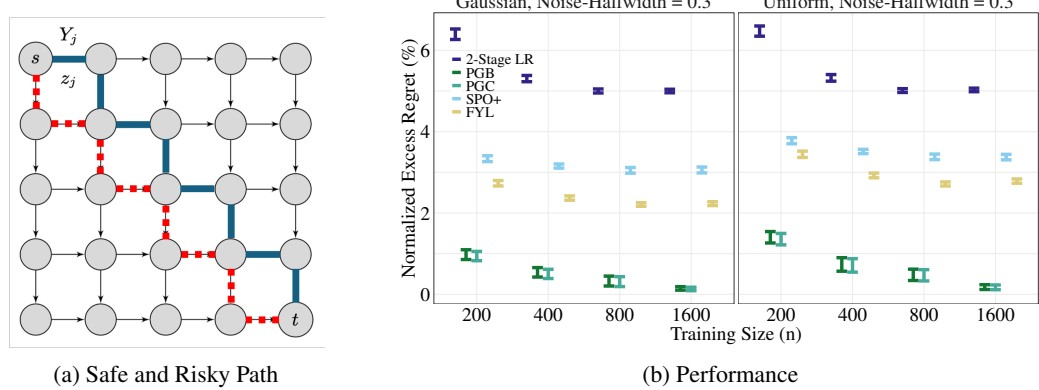

(a) Safe and Risky Path        (b) Performance

Figure 4: Harder Shortest Path. a) One of the two planted paths will be optimal depending on value of $X_6$. All other arcs strictly worse. b) Normalized Excess Regret as we vary the training samples. Error bars are 95% confidence intervals on the mean over 100 trials.

Specifically, we now take $X \in \mathbb{R}^6$ where $X_{1:5} \sim \mathcal{N}(0, \boldsymbol{I}_5)$ and $X_6 \sim$ Unif$[0, 2]$. In Fig. 4a, we have a safe (red) path and a risky (blue) path. For red arcs, $f_j^*(x) = 2$ for all $x$. For the blue arcs (risky

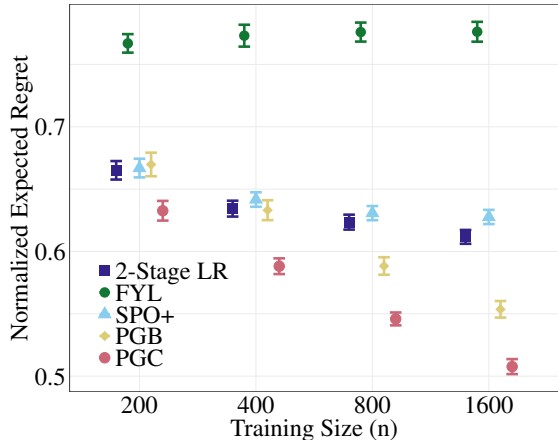

Figure 5: (Portfolio Optimization) We plot the excess regret normalized by optimal policy's performance as we vary the number of training samples. Error bars are 95% confidence intervals on the mean over 100 trials.

path), $f_j^*(x) = 4x_6$ if $0 \le x_6 \le 0.55$ and $f_j^*(x_6) = 2.2$ otherwise. For all other arcs, we take

$$f_j^*(x) = \frac{1}{3.5^6}\left[\left(\frac{1}{\sqrt{5}}(B^*x)_j + 3\right)^6 + 1\right] + 2.2,$$

which is similar to previous experiment but shifted up by 2.2. Consequently, either the red path or the blue path is optimal, depending on the value of $X_6$. The observed $Y$ values are generated as before by adding either multiplicative uniform or additive Gaussian noise. A good method thus must first identify these two paths as the best options (despite the noise) and choose between them (by learning the relationship to $X_6$). In this harder setting, PG losses offer a significant benefit. Figure 9 in Appendix D shows this performance is relatively robust to the choice of $h$.

### 4.3 Portfolio Experiment

We study the same portfolio optimization problem as [7, 28, 34] but use real data, specifically the 12 Fama French Industry Sector Portfolios from the Fama French Library [9]. These portfolios are indices representing monthly returns of different asset classes and realistically mirror the asset allocation problem faced by wealth managers. We sample a month $t$ at random from the last 10 years, and let $Y = r_t$ be the return of the $d = 12$ indices, and let $X = r_{t-1} + \mathcal{N}(0, 0.5\Sigma)$ $(p = 12)$ where $\Sigma$ is the covariance of $r_t$ over those 10 years. The additional noise lowers the signal-to-noise ratio while maintaining the correlation matrix of $X$ and makes the problem harder.

As one can see in Fig. 5, because of the low signal-to-noise ratio, all methods induce significant regret to the optimum, but both PGB and PGC are notably stronger.

## 5 Conclusion

In this paper we proposed a novel family of surrogate losses for the predict-then-optimize framework that can be optimized using off-the-shelf gradient methods. Most importantly, the approximation error of these surrogates vanishes as $n \to \infty$. Hence, optimizing our surrogate yields a best-in-class policy asymptotically, even in misspecified settings. Our PG losses are the first proposed surrogates with this property and substantively outperform other methods in misspecified settings.

The family of PG losses arises from different approaches to approximating a derivative. As mentioned, an interesting open question is identifying the best-possible choice of approximation. We also believe that better understanding the role of $h$ in trading off between bias and computational complexity might shed light on improve algorithms and tuning procedures.

## Acknowledgments and Disclosure of Funding

The authors have no competing interests to disclose. The authors would like to thank Hamsa Bastani, Osbert Bastani, Adam Elmachtoub, Paul Grigas, Ziyu He, and Angela Zhou for feedback on an initial draft of this manuscript. VG was partially funded by the Institute for Outlier Research in Business.

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

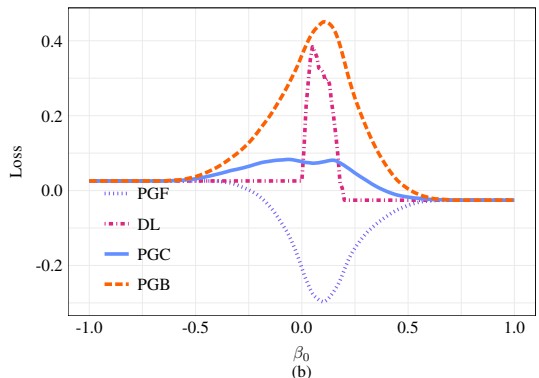

Figure 6: (Comparing Zeroth Order Gradients). PGC, PGB, and PGF all approximate the decision loss (DL), but PGB is a pessimistic bound, while PGF is an optimistic bound. Here the optimism causes PGF to choose the wrong policy.

## Appendix / Supplemental Material

## A    Reformulating Nonlinear Problems

Through an appropriate transfomation of variables, some nonlinear optimization problems can be rewritten in the form Problem (1), and, thus, are amenable to our approach.

Consider the problem

$$\pi^*(X) \in \operatorname*{argmin}_{z \in \mathcal{Z}} f^*(X)^\top g(z),$$

where $f^*(X) = \mathbb{E}\left[Y \mid X\right]$ and $g(\cdot)$ is a fixed, known, vector-valued function. This problem is equivalent to the problem

$$\min_{\bar{z}} \quad f^*(X)^\top \bar{z}$$
$$\text{s.t.} \quad \bar{z} \in \bar{\mathcal{Z}} \equiv \{g(z) : z \in \mathcal{Z}\},$$

which is of the requisite form for our analysis. Moreover, our algorithms only require access to an oracle which can compute $\bar{\pi}(f(X)) \in \operatorname{argmin}_{\bar{z} \in \bar{\mathcal{Z}}} f(X)^\top \bar{z}$ for any $f$. Often, this is accomplished by solving $\pi(f(X)) \in \operatorname{argmin}_{z \in \mathcal{Z}} f(X)^\top z$ and then returning $g(\pi(f(X)))$.

Gupta, Huang, and Rusmevichientong [12] use this reduction to model a personalized pricing problem (see Example 2.3 of their paper).

## B    Comparing Zeroth Order Gradient Schemes

In this section we provide a brief comparison of the forward differencing scheme to backwards and central differencing. The key distinction is that since $V(\cdot)$ is concave, forward differencing creates a surrogate that optimistically underestimates the true loss (forward differences underestimate the derivative of concave functions) whereas backward differencing pessimistically overestimates the true loss. Some authors [4, 33] have shown that pessimism can improve learning, and we observe a similar phenomenon.

Figure 6 provides an illustration. We consider the same misspecified data setup as Section 4.1 ($\alpha = 1$, $m = 0$) and take $n = 200$. We plot the decision loss (DL) $\ell$, and our PGB, PGF, and PGC losses, for the plug-in class $\mathcal{F} = \{-0.1x + B_0 : B_0 \in [-1, 1]\}$. Because PGF optimisticaly underestimates loss, it suggests the policy $\beta_0 = .1$, which actually induces significant regret. By contrast, backwards differencing is pessimistic and suggests the policy $\beta_0 = .98$ which is essentially optimal. Central differencing is neither optimistic nor pessimistic, but still suggests a good policy $\beta_0 = .99$.

## C    Implementation Details

For our numerical experiments we leverage the PyEPO framework which was developed using PyTorch. For our experiments, we utilize Adam with learning rate 0.01 to optimize the training losses. We run Adam over 100 epochs with a batch size of 32 for each surrogate loss. For non-PG loss

surrogates we use the recommended parameters provided by PyEPO. For our PG losses, we tune $h$ by validating with a hold out set of training 200 samples. We note that similar results were obtained by validating against the training decision loss. Additionally, we initialize the PG losses at the SPO+ solution.

To compute the expected regret, we generated a test set of 10000 samples and use it to estimate the relative regret described in Section 4.

Some of our experiments were run on a high performance computing cluster administred by the University of Southern California's Center for Advanced Research Computing (CARC). The cluster facilitated multiple simulation runs of the experiments. However, a significant portion of the experiments in the paper (that did not require multiple Monte Carlo runs) were run on a Macbook Pro with an Apple M3 Max Chip with 96 GB Memory.

# D    Additional Figures

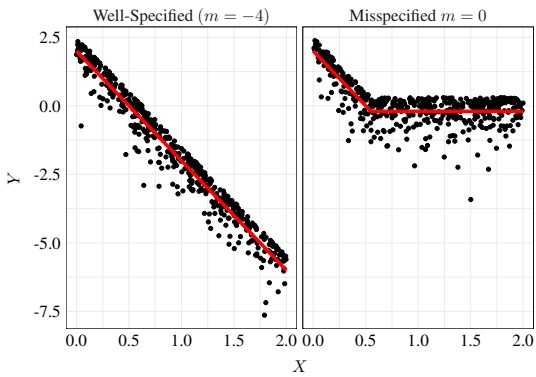

Figure 7: (Synthetic Data Generation from Section 4.1) Observations of $(X_i, Y_i)$ for $m = -4$ (left) and $m = 0$ (right). Red line is $f^*(X)$ for each setting.

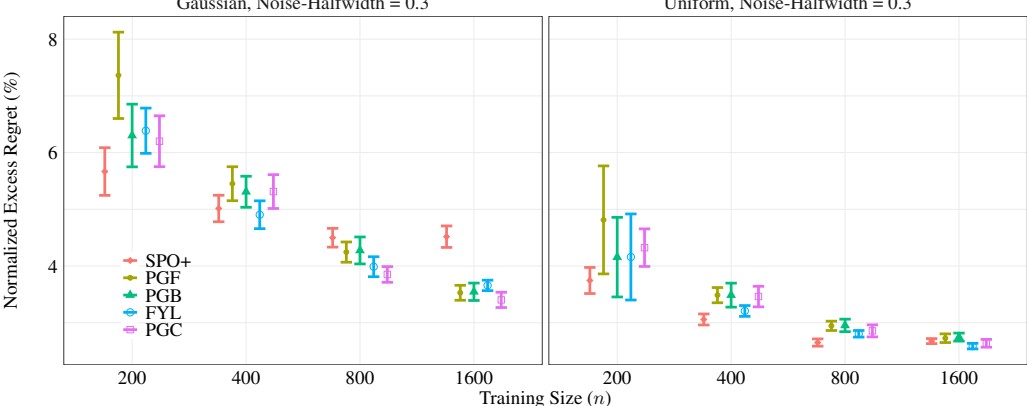

Figure 8: (Shortest Path, Random Arc Costs) Excess regret normalized by optimal policy's performance as we vary the number of training samples. Error bars are $95\%$ confidence intervals on the mean over 100 trials.

# E    Omitted Proofs

## E.1    Proof for Lemma 2.1

*Proof.* We first prove (a), the Lipschitz property. We first claim $V(\cdot)$ is $B$ Lipschitz, since

$$V(t) - V(s) = t^\top \hat{\pi}(t) - s^\top \hat{\pi}(s) = \underbrace{t^\top (\hat{\pi}(t) - \hat{\pi}(s))}_{\leq 0, \text{ by optimality of } \hat{\pi}(t)} + (t - s)^\top \hat{\pi}(s)$$

$$\leq \|t - s\| \|\hat{\pi}(s)\| \leq B \|t - s\|,$$

| n | ep_type | h | avg | std |
|---|---|---|---|---|
| 800 | normal | 0.001 | 0.050 | 0.008 |
| 800 | normal | 0.035 | 0.049 | 0.008 |
| 800 | normal | 0.188 | 0.048 | 0.008 |
| 800 | normal | 0.434 | 0.049 | 0.013 |
| 800 | unif | 0.001 | 0.074 | 0.009 |
| 800 | unif | 0.035 | 0.073 | 0.009 |
| 800 | unif | 0.188 | 0.072 | 0.010 |
| 800 | unif | 0.434 | 0.076 | 0.017 |
| 1600 | normal | 0.001 | 0.048 | 0.005 |
| 1600 | normal | 0.025 | 0.047 | 0.004 |
| 1600 | normal | 0.158 | 0.046 | 0.004 |
| 1600 | normal | 0.398 | 0.048 | 0.012 |
| 1600 | unif | 0.001 | 0.070 | 0.006 |
| 1600 | unif | 0.025 | 0.070 | 0.006 |
| 1600 | unif | 0.158 | 0.068 | 0.006 |
| 1600 | unif | 0.398 | 0.071 | 0.015 |

(a) Validation Performance

| n | ep_type | h | avg | std |
|---|---|---|---|---|
| 800 | normal | 0.001 | 0.006 | 0.006 |
| 800 | normal | 0.035 | 0.005 | 0.006 |
| 800 | normal | 0.188 | 0.004 | 0.006 |
| 800 | normal | 0.434 | 0.005 | 0.011 |
| 800 | unif | 0.001 | 0.007 | 0.007 |
| 800 | unif | 0.035 | 0.006 | 0.007 |
| 800 | unif | 0.188 | 0.006 | 0.008 |
| 800 | unif | 0.434 | 0.009 | 0.015 |
| 1600 | normal | 0.001 | 0.003 | 0.002 |
| 1600 | normal | 0.025 | 0.003 | 0.002 |
| 1600 | normal | 0.158 | 0.002 | 0.002 |
| 1600 | normal | 0.398 | 0.003 | 0.010 |
| 1600 | unif | 0.001 | 0.004 | 0.003 |
| 1600 | unif | 0.025 | 0.004 | 0.003 |
| 1600 | unif | 0.158 | 0.002 | 0.003 |
| 1600 | unif | 0.398 | 0.005 | 0.011 |

(b) Normalized Excess Regret

Figure 9: (Dependence on $h$, Planted Shortest Path Experiment.) We compare the performance of the policy learned by the PGB loss for different values of $h$ across the 100 runs. a) Shows performance on Valiation set. For ease of comparison, we have scaled the validation performance and presented $\left(\sum_{i=1}^{n_{val}} Y_i^\top (\hat{\pi}(f(X_i)) - \hat{\pi}(f^*(X_i)))\right) / \left(\sum_{i=1}^{n_{val}} Y_i^\top \hat{\pi}(f^*(X_i))\right)$. b) Shows performance out of sample. This out-of-sample performance is relatively flat in $h$, suggesting the precise choice of $h$ does not matter much in this example.

where the last inequality follows from Assumption 1.1. A symmetric argument holds for $V(s) - V(t)$ proving $V$ is $B$ Lipschitz.

Returning to $\hat{\ell}_h^b(t, y)$, write

$$
\begin{aligned}
\left|\hat{\ell}_h^b(t, y) - \hat{\ell}_h^b(t', y)\right| &= \left|\frac{V(t) - V(t - hy)}{h} - \frac{V(t') - V(t' - hy)}{h}\right| \\
&\leq \frac{|V(t) - V(t')|}{h} + \frac{|V(t' - hy) - V(t - hy)|}{h} \\
&\leq \frac{2B|t - t'|}{h}.
\end{aligned}
$$

An entirely analogous argument holds for $\hat{\ell}_h^c(t, y)$.

We next prove (b), the boundedness property. Write

$$
\left|\hat{\ell}_h^b(t, y)\right| = \frac{|V(t) - V(t - hy)|}{h} \leq \frac{B\|hy\|}{h} = B\|y\|
$$

Again, an analogous argument holds for $\hat{\ell}_h^c(t, y)$. This completes the proof for (b)

The proof of (c) follows directly from applying Danskin's Theorem [2, Prop B.22].

To prove (d), we see

$$
\begin{aligned}
\hat{\ell}_h^b(t, y) - \ell(t, y) &= \frac{V(t + hy) - V(t)}{h} - y^\top \hat{\pi}(t) \\
&= \frac{(t + hy)^\top \hat{\pi}(t + hy) - (t + hy)^\top \hat{\pi}(t)}{h} \\
&\geq 0
\end{aligned}
$$

where the last inequality holds by optimality of $\hat{\pi}(t + hy)$. Rearranging proves the result for (d). □

## E.2 Proof of Lemma 2.2.

*Proof.* We apply the dominated convergence theorem. Let $e_i \in \mathbb{R}^d$ be the $i^{\text{th}}$ coordinate vector. Then,

$$\partial_{t_i} \mathbb{E}\left[\hat{\ell}_h^b(t, Y)\right] = \lim_{\delta \to 0} \mathbb{E}\left[\frac{1}{\delta}(\hat{\ell}_h^b(t + \delta, Y) - \hat{\ell}_h^b(t, Y))\right] \tag{6}$$

Let $W_\delta \equiv \frac{1}{\delta}(\hat{\ell}_h^b(t + \delta, Y) - \hat{\ell}_h^b(t, Y))$. Then, by the Lipschitz property of Lemma 2.1, $|W_\delta| \leq \frac{2B}{h}$, and $\lim_{\delta \to 0} W_\delta = \partial_{t_i} \hat{\ell}_h^b(t, Y)$ almost surely. The result then holds for the $i^{\text{th}}$ partial derivative of $\hat{\ell}_h^b$ from the dominated convergence theorem. Since $i$ was arbitrary, it holds for all $i = 1, \ldots, d$, and thus holds for the gradient. An analogous proof holds for $\hat{\ell}_h^c$. $\square$

## E.3 Auxiliary Lemmas from Section 3

**Lemma E.1** (Interchange Derivative for $H$)**.** *Suppose Assumption 1.1 holds and that the optimizer $\hat{\pi}(T + \lambda Y)$ is unique almost surely. Then $H'(\lambda) = \mathbb{E}\left[\frac{d}{d\lambda} V(T + \lambda Y)\right]$.*

*Proof.* We use the bounded convergence theorem. Write

$$H'(\lambda) = \lim_{\delta \to 0} \frac{H(\lambda + \delta) - H(\lambda)}{\delta}$$
$$= \lim_{\delta \to 0} \mathbb{E}\left[\frac{V(T + (\lambda + \delta)Y) - V(T + \lambda Y)}{\delta}\right].$$

Because $V$ is $B$-Lipschitz, $\left|\frac{V(T+(\lambda+\delta)Y)-V(T+\lambda Y)}{\delta}\right| \leq \|Y\| \leq 1$. By the bounded convergence theorem we can interchange the limit and expectation yielding,

$$H'(\lambda) = \mathbb{E}\left[\lim_{\delta \to 0} \frac{V(T + (\lambda + \delta)Y) - V(T + \lambda Y)}{\delta}\right].$$

Since $\hat{\pi}(T + \lambda Y)$ is unique, Danskin's theorem [2, Prop B.22] confirms $V(T + \lambda Y)$ is differentiable, and the above inner limit converges to the derivative $\frac{d}{d\lambda} V(T + \lambda Y)$. $\square$

**Lemma E.2** (Error of Backward Finite Difference)**.** *Suppose $H$ is differentiable on $[\lambda - h, \lambda]$, and $\beta$-smooth. Then,*

$$\left|H'(\lambda) - \frac{1}{h}(H(\lambda) - H(\lambda - h))\right| \leq \beta h.$$

*Proof.* By the mean-value theorem, $\frac{1}{h}(H(\lambda) - H(\lambda - h)) = H'(\lambda - \bar{h})$ for some $0 \leq \bar{h} \leq h$. Thus,

$$\left|H'(\lambda) - \frac{1}{h}(H(\lambda) - H(\lambda - h))\right| = \left|H'(\lambda) - H'(\lambda - \bar{h})\right| \leq \beta \bar{h},$$

by $\beta$-smoothness. Upper bounding $\bar{h}$ by $h$ completes the proof. $\square$

## E.4 Proof for Lemma 3.2

Our first observation is that the error in our surrogate is bounded by the solution stability of the policy. A similar bound is used in Gupta, Huang, and Rusmevichientong [13] in a different context:

**Lemma E.3** (Solution Stability Bounds Error)**.** *For any $t, y, h$,*

$$0 \leq \hat{\ell}_h^b(t, y) - \ell(t, y) \leq \underbrace{y^\top \left(\hat{\pi}(t - hy) - \hat{\pi}(t)\right)}_{\text{Solution Stability}}$$

In words, solution stability measures how much the policy changes given small perturbation $hy$. Notions of stability appear throughout the machine learning literature and are fundamental to learnability [29]. Lemma E.3 relates the error of our surrogate to this fundamental quantity. We stress the relation holds for *any* $t, h, y$.

*Proof.* The first inequality was proven in Lemma 2.1. For the second, note that $V(t) = t^\top \hat{\pi}(t)$. Hence by rearranging,

$$\hat{\ell}_h^b(t, y) - \ell(t, y) = \frac{1}{h}(V(t) - V(t - hy)) - y^\top \hat{\pi}(t)$$
$$= \frac{1}{h}\left(t^\top(\hat{\pi}(t) - \hat{\pi}(t - hy)) + y^\top(\hat{\pi}(t - hy) - \hat{\pi}(t))\right)$$
$$\leq y^\top(\hat{\pi}(t - hy) - \pi(t)),$$

by the optimality of $\hat{\pi}(t)$. $\qquad\square$

To bound the expected approximation error in Lemma 3.2, we require the following elementary result:

**Lemma E.4** (Density Ratio Bound). *Suppose Assumption 3.1 holds. Then, for any $t, t'$ such that $\|t - t'\| \leq 1/L$, we have*

$$\left|\frac{g(t'; f, Y)}{g(t; f, Y)} - 1\right| \leq (e - 1)L\|t - t'\|.$$

*Proof.* Let $g(t) \equiv g(t; f, Y)$. By the convexity of the exponential,

$$\exp(x) \leq 1 + (e - 1)x \ \ \forall 0 \leq x \leq 1, \quad \text{and} \quad \exp(x) \geq 1 + x \ \ \forall x. \tag{7}$$

Let $s(t) = \log g(t)$. Then,

$$\log\left(\frac{g(t')}{g(t)}\right) = s(t') - s(t) \leq L\|t' - t\|$$

Taking the exponential of both sides and subtracting 1, we have

$$\frac{g(t')}{g(t)} - 1 \leq \exp\left(L\|t' - t\|\right) - 1$$
$$\leq (e - 1)L\|t' - t\|,$$

where the last inequality follows from Eq. (7) and our assumption that $\|t - t'\| \leq 1/L$. Similarly, we have,

$$\log\left(\frac{g(t')}{g(t)}\right) \geq -L\|t' - t\|$$
$$\frac{g(t')}{g(t)} - 1 \geq \exp\left(-L\|t' - t\|\right) - 1$$
$$\geq -L\|t' - t\|$$
$$\geq -(e - 1)L\|t' - t\|$$

Hence,

$$\left|\frac{g(t')}{g(t)} - 1\right| \leq (e - 1)L\|t' - t\|.$$

This completes the proof. $\qquad\square$

*Proof of Lemma 3.2.* Let $T = f(X)$. Condition on $Y$ and let $g(t) \equiv g(t; f, Y)$. Then, by Lemma E.3, we have

$$0 \leq \mathbb{E}\left[\hat{\ell}_h^b(T, Y) - \ell(T, Y)\Big| Y\right] \leq \mathbb{E}\left[Y^\top(\hat{\pi}(T - hY) - \hat{\pi}(T))\Big| Y\right].$$

We bound this last quantity as follows:

$$\mathbb{E}\left[Y^\top \left(\hat{\pi}(T - hY) - \hat{\pi}(T)\right) \big| Y\right] \tag{8}$$

$$= \int g(t) Y^\top \hat{\pi}(t - hY) dt - \int g(t) Y^\top \hat{\pi}(t) dt$$

$$= \int Y^\top \hat{\pi}(t) \left(g(t + hY) - g(t)\right) dt$$

$$\leq \int \left|Y^\top \hat{\pi}(t)\right| \left|g(t + hY) - g(t)\right| dt$$

$$\leq B \int g(t) \left|\frac{g(t + hY)}{g(t)} - 1\right| dt$$

$$\leq (e - 1) BL \|hY\| \int g(t; f, Y) dt$$

$$\leq (e - 1) BLh$$

Taking the expectation over $Y$ completes the proof. $\qquad\square$

## E.5  Proof for Theorem 3.4

*Proof.* We bound the uniform error as follows:

$$\sup_{f \in \mathcal{F}} \left|\frac{1}{n} \sum_{i=1}^n \hat{\ell}_h^b\left(f(X_i), Y_i\right) - \mathbb{E}\left[\ell\left(f(X_i), Y_i\right)\right]\right| \leq \underbrace{\sup_{f \in \mathcal{F}} \left|\frac{1}{n} \sum_{i=1}^n \hat{\ell}_h^b\left(f(X_i), Y_i\right) - \mathbb{E}\left[\hat{\ell}_h^b\left(f(X_i), Y_i\right)\right]\right|}_{(i)}$$

$$+ \underbrace{\sup_{f \in \mathcal{F}} \left|\frac{1}{n} \sum_{i=1}^n \mathbb{E}\left[\hat{\ell}_h^b\left(f(X_i), Y_i\right) - \ell\left(f(X_i), Y_i\right)\right]\right|}_{(ii)}$$

We first bound $(i)$. Let

$$\mathfrak{R}_{SL}^n(\mathcal{F}) = \mathbb{E}\left[\hat{\mathfrak{R}}_{SL}^n(\mathcal{F})\right] = \mathbb{E}\left[\mathbb{E}_\sigma\left[\sup_{f \in \mathcal{F}} \frac{1}{n} \sum_{i=1}^n \sigma_i \hat{\ell}_h^b\left(f(X_i), Y_i\right)\right]\right].$$

By Lemma 2.1b, $0 \leq \frac{\hat{\ell}_h^b(f(X_i), Y_i) + B}{2B} \leq 1$. Hence, we can apply a standard Rademacher complexity result [23, Theorem 3.3] to show for any $0 < \delta < \frac{1}{2}$, with probability at least $1 - \delta$, the following holds for all $f \in \mathcal{F}$ simultaneously:

$$\frac{1}{n} \sum_{i=1}^n \mathbb{E}\left[\frac{\hat{\ell}_h^b(f(X_i), Y_i) + B}{2B}\right] \leq \frac{1}{n} \sum_{i=1}^n \frac{\hat{\ell}_h^b(f(X_i), Y_i) + B}{2B} + 2\mathfrak{R}_{SL}^n(\mathcal{F}) + \sqrt{\frac{1}{n} \log\left(\frac{1}{\delta}\right)}.$$

We can apply an identical argument to $\frac{-\hat{\ell}_h^b(f(X_i), Y_i) + B}{2B}$ to obtain a similar lower bound. Combining the two inequalities and taking the union bound, we have that with probability at least $1 - 2\delta$, the following holds for all $f \in \mathcal{F}$ simultaneously:

$$\left|\frac{1}{n} \sum_{i=1}^n \hat{\ell}_h^b\left(f(X_i), Y_i\right) - \mathbb{E}\left[\hat{\ell}_h^b\left(f(X_i), Y_i\right)\right]\right| \leq 4B\mathfrak{R}_{SL}^n(\mathcal{F}) + 2B\sqrt{\frac{1}{n} \log\left(\frac{1}{\delta}\right)}$$

We next bound $\mathfrak{R}_{SL}^n(\mathcal{F})$ by applying Corollary 4 of Maurer [22] to show

$$\mathfrak{R}_{SL}^n(\mathcal{F}) = \mathbb{E}\left[\sup_{f \in \mathcal{F}} \frac{1}{n} \sum_{i=1}^n \sigma_i \hat{\ell}_h^b\left(f(X_i), Y_i\right)\right] \leq \sqrt{2}\frac{B}{h}\mathbb{E}\left[\sup_{f \in \mathcal{F}} \frac{1}{n} \sum_{i=1}^n \sigma_i^\top f(X_i)\right] = \sqrt{2}\frac{B}{h}\mathfrak{R}^n(\mathcal{F}).$$

Here we have used the Lipschitz constant from Lemma 2.1a.

Substituting this bound above and collecting constants shows that with probability at least $1 - \delta$,

$$(i) \;\lesssim\; \frac{B^2}{h}\mathfrak{R}^n(\mathcal{F}) + B\sqrt{\frac{\log(1/\delta)}{n}}.$$

Finally, we use Lemma 3.2 to bound $(ii)$. Combining proves the result. $\qquad\square$

## E.6 Proof for Theorem 3.7

*Proof.* We develop an alternative decomposition of the uniform error. Write

$$\left|\frac{1}{n}\sum_{i=1}^{n}\hat{\ell}_h^b\left(f(X_i), Y_i\right) - \mathbb{E}\left[\ell\left(f(X_i), Y_i\right)\right]\right| \;\leq\; \underbrace{\left|\frac{1}{n}\sum_{i=1}^{n}\hat{\ell}_h^b\left(f(X_i), Y_i\right) - \ell\left(f(X_i), Y_i\right)\right|}_{(i)} \tag{9}$$

$$+ \left|\frac{1}{n}\sum_{i=1}^{n}\ell\left(f(X_i), Y_i\right) - \mathbb{E}\left[\ell\left(f(X_i), Y_i\right)\right]\right|$$

Consider $(i)$. We can write

$$(i) \;\leq\; \frac{1}{n}\sum_{i=1}^{n}\left|\hat{\ell}_h^b\left(f(X_i), Y_i\right) - \ell\left(f(X_i), Y_i\right)\right|$$

$$\leq\; \frac{1}{n}\sum_{i=1}^{n} Y_i^\top \left(\hat{\pi}(f(X_i) - hY_i) - \hat{\pi}(f(X_i))\right)$$

$$=\; \frac{1}{n}\sum_{i=1}^{n} Y_i^\top \hat{\pi}(f(X_i) - hY_i) - \mathbb{E}\left[Y_i^\top \hat{\pi}(f(X_i) - hY_i)\right]$$

$$-\; \frac{1}{n}\sum_{i=1}^{n} Y_i^\top \hat{\pi}(f(X_i)) - \mathbb{E}\left[Y_i^\top \hat{\pi}(f(X_i))\right]$$

$$+\; \frac{1}{n}\sum_{i=1}^{n} \mathbb{E}\left[Y_i^\top \left(\hat{\pi}(f(X_i) - hY_i) - \hat{\pi}(f(X_i))\right)\right]$$

$$\leq\; 2\sup_h \left|\frac{1}{n}\sum_{i=1}^{n} Y_i^\top \hat{\pi}(f(X_i) - hY_i) - \mathbb{E}\left[Y_i^\top \hat{\pi}(f(X_i) - hY_i)\right]\right|$$

$$+\; \frac{1}{n}\sum_{i=1}^{n} \mathbb{E}\left[Y_i^\top \left(\hat{\pi}(f(X_i) - hY_i) - \hat{\pi}(f(X_i))\right)\right]$$

where the first inequality applies the triangle inequality, the second inequality applies Lemma E.3, and the last inequality combines similar terms by taking the supremum over $h$.

Applying this bound in Eq. (9) shows

$$\sup_{f\in\mathcal{F}}\left|\frac{1}{n}\sum_{i=1}^{n}\hat{\ell}_h^b\left(f(X_i), Y_i\right) - \mathbb{E}\left[\ell\left(f(X_i), Y_i\right)\right]\right|$$

$$\leq\; 3\sup_{\bar{f}\in\bar{\mathcal{F}}}\underbrace{\left|\frac{1}{n}\sum_{i=1}^{n} Y_i^\top \hat{\pi}(\bar{f}(X_i, Y_i)) - \mathbb{E}\left[Y_i^\top \hat{\pi}(\bar{f}(X_i, Y_i))\right]\right|}_{(a)}$$

$$+\; \sup_{f\in\mathcal{F}}\underbrace{\frac{1}{n}\sum_{i=1}^{n} \mathbb{E}\left[Y_i^\top \left(\hat{\pi}(f(X_i) - hY_i) - \hat{\pi}(f(X_i))\right)\right]}_{(b)}$$

where we recall that

$$\bar{\mathcal{F}} = \left\{\bar{f} : \bar{f}(x, y) = f(x) + hy, \text{ for } f \in \mathcal{F}, h \in \mathbb{R}\right\}.$$

Component $(a)$ is bounded using Theorem 1 and Theorem 2 of Hu, Kallus, and Mao [14] showing that with probability at least $1 - \delta$,

$$(a) \lesssim B\sqrt{\frac{\nu \log\left(|\mathcal{Z}_\angle| + 1\right)\log(5/\delta)}{n}}.$$

Component $(b)$ is bounded by Eq. (8) in the proof of Lemma 3.2. Combining $(a)$ and $(b)$ components proves the result. $\qquad\square$

### E.7 Proof of Theorem 3.8

*Proof.* Both proofs follow the same general strategy. We start with the first statement. Let

$$L_n(f) = \frac{1}{n}\sum_{i=1}^{n} \hat{\ell}_h^b\left(f(X_i), Y_i\right) \quad \text{and} \quad L(f) = \mathbb{E}\left[\ell\left(f(X), Y\right)\right]$$

Since the $\hat{f}_b$ minimizes $L_n(f)$ over $\mathcal{F}$ and $f^{OR}$ minimizes $L(f)$ over $\mathcal{F}$, we see,

$$
\begin{aligned}
L(\hat{f}_b) - L(f^{OR}) &= L(\hat{f}_b) - L_n(\hat{f}) + L_n(\hat{f}_b) - L_n(f^{OR}) + L_n(f^{OR}) - L(f^{OR}) \\
&\leq \underbrace{L_n(\hat{f}) - L_n(f^{OR})}_{\leq 0,\ \text{by optimality of } \hat{f}} + 2\sup_{f\in\mathcal{F}}|L_n(f) - L(f)| \\
&\leq 2\sup_{f\in\mathcal{F}}|L_n(f) - L(f)|
\end{aligned}
$$

where the first inequality holds by taking the supremum of the first two and last two pairs, and the second inequality holds by optimality of $\hat{f}$. Taking the expectation of both sides, we see

$$\mathrm{ERegret}(\hat{f}_b) \leq 2\mathbb{E}\left[\sup_{f\in\mathcal{F}}|L_n(f) - L(f)|\right]$$

To compute the expectation, we see by Theorem 3.4 and choosing $h = \sqrt{\frac{B}{L}\mathfrak{R}^n(\mathcal{F})}$ that

$$\sup_{f\in\mathcal{F}}|L_n(f) - L(f)| \leq \sqrt{B^3 L\mathfrak{R}^n(\mathcal{F})} + B\sqrt{\frac{1}{n}\log\frac{1}{\delta}}. \tag{10}$$

with probability at least $1 - \delta$. Rearranging, we have

$$\mathbb{P}\left(\sup_{f\in\mathcal{F}}|L_n(f) - L(f)| - \sqrt{B^3 L\mathfrak{R}^n(\mathcal{F})} \geq t\right) \leq \exp\left(-\frac{nt^2}{B^2}\right)$$

By tail integration over $t$ and adding back $\sqrt{B^3 L\mathfrak{R}^n(\mathcal{F})}$, it follows that

$$\mathrm{ERegret}(\hat{f}_b) \leq 2\mathbb{E}\left[\sup_{f\in\mathcal{F}}|L_n(f) - L(f)|\right] \lesssim \sqrt{B^3 L\mathfrak{R}^n(\mathcal{F})} + \frac{B}{\sqrt{n}},$$

completing the proof of the first statement.

We now proceed to the second statement. We follow the same line of argument until Eq. (10). Then, we instead use Theorem 3.7 with $h = \frac{1}{L\sqrt{n}} \leq \frac{1}{L}$ to obtain

$$\sup_{f\in\mathcal{F}}|L_n(f) - L(f)| \leq C'B\sqrt{\frac{\nu \log\left(|\mathcal{Z}_\angle| + 1\right)\log(1/\delta)}{n}}$$

for some universal constant $C_0$ with probability at least $1 - \delta$. Rearranging we have

$$\mathbb{P}\left(\sup_{f\in\mathcal{F}}|L_n(f) - L(f)| \geq t\right) \leq \exp\left(-\frac{nt^2}{C_0^2 B^2 \nu \log\left(|\mathcal{Z}_\angle| + 1\right)}\right).$$

Applying the tail integral gives us

$$\mathrm{ERegret}(\hat{f}_b) \leq 2\mathbb{E}\left[\sup_{f\in\mathcal{F}}|L_n(f) - L(f)|\right] \lesssim B\sqrt{\frac{\nu \log\left(|\mathcal{Z}_\angle| + 1\right)}{n}}$$

completing the proof.

$\qquad\square$

