# OpenReview forum: "Decision-Focused Learning with Directional Gradients"
_NeurIPS.cc/2024/Conference — NeurIPS 2024 poster_

### Official Review · Reviewer_oPp7 · 2024-07-07

**Soundness:** 2
**Presentation:** 2
**Contribution:** 3
**Rating:** 5
**Confidence:** 3

**Summary:**

The paper introduces a new family of surrogate losses to the DFL with linear costs, called perturbation gradient losses (PG loss). It provides theoretical analysis to bound the approximation errors and regret bounds and uses extensive experiments to demonstrate the advantages of the proposed method.

**Strengths:**

- The proposed loss is more efficient than the standard DFO loss. It only requires the computation of the optimal decision variable without the need for implicit differentiation through $\hat{\pi}(t)$.
- The approximation error of the surrogate functions decreases as the number of samples increases, which is advantageous for large-sample applications.
- In the numerical experiment section, the PG losses perform very well even in the misspecification case.

**Weaknesses:**

- The (sub)differentiability of the PG loss is not well-established. In Line 172, the differentiability is doubtful. I don’t think the subgradient is well defined when $\hat{\pi}$ is not unique. The paper should make more efforts to develop the theory rigorously and in writing.
- In Line 148, the directional derivative is not well-defined. If $V(t)$ is non-smooth, you should take infimum over all the $\hat{\pi}(t)$.
- The paper presents a continuous relaxation of the DFL objective, with the error bounded by the finite difference error h. However, the resulting problem is still nonconvex and nonsmooth, which may not be efficiently solved in theory. It seems more effective to use direct smoothing by random perturbation or by adding a regularization term (e.g., [30]). I encourage the authors to compare their approach with other smoothing methods in theory.
- Both experiments are for discrete decision variables and simulated data.  The paper does not sufficiently address the performance of the proposed method when decisions are continuous.

**Questions:**

- Lemma 2.2 Can you develop the result when $\ell_h^b$ is nonsmooth?
- Line 176, what is $Y_j$?

---

> ### Author Rebuttal · Authors · 2024-08-06
>
> ### **Danskin's Theorem**
> There are many versions of Danskin’s Theorem that apply under different regularity conditions. The version we use is well-summarized [here](https://statisticaloddsandends.wordpress.com/2022/11/10/what-is-danskins-theorem/).  The key is part 4:  When there are multiple solutions, the theorem holds but with “gradient” replaced by “subgradient.” An intuitive example tis $f(t) = \max_{-1 \leq z \leq 1} t z = | t |$. Then, at $t = 0$, the set of optimal solutions is $[-1, 1]$ (non-unique), and thus no derivative. Still, there are subgradients, and any optimal solution gives a subgradient.
>
> Given your excellent questions we intend to add a statement of Danskin’s Theorem at the end of Section 1.3.
>
> ### **Addressing Weaknesses (1-2)**
> 1. As in part 4 of reference above, Danskin’s theorem holds when $\hat \pi(\cdot)$ is not unique. Since $\hat \pi(\cdot)$ is AN optimal solution, it is a subgradient of V(t).  See also footnote 1 of pg. 5 of our paper.
>
> 2. Again, we are invoking part 4.  The “grad”  operator should be interpreted as a directional gradient if
> $\hat \pi(t)$ is unique and a Clarke Differential otherwise.
>
> ### **Addressing Weakness 3: Comparison to Smoothing and Regularization**
> #### ***NP Hardness***
> Optimizing decision-loss (DL) generalizes binary classification and is NP-Hard [6]. Hence, ANY method (including smoothing and regularization) that aims to learn a best-in-class policy must either suffer similar computational challenges as our method or else fail to recover a best-in-class policy.
>
> #### ***Regularization [30]***  In [30], the authors
> i) relax combinatorial constraints,
> ii) smooth by introducing a regularized policy class $\hat \pi^\rho(t) \in \arg\min_{z \in \mathcal Z} \langle t, z \rangle + \rho || z ||^2$
> iii) approximately optimize the decision-loss over these regularized policies
> iv) implement the non-regularized $\hat \pi(t)$ using the parameters learned from the regularized $\hat \pi^\rho(t)$.
>
> They provide no theoretical analysis of the approximation error. (For combinatorial problems, intuition suggests the error from the first step might be very large.) Even in very simple settings -- take binary classification $\mathcal Z = [-1, 1]$, $Y \in \{-1, 1\}$ --  their loss can have large flat regions and is non-convex. Hence step iii) does not appear (to us) to be any less challenging than our optimization problem.
>
> The method of [30] is also computationally expensive. Evaluating gradients in step iii) entails solving a QP. By contrast, our method only involves solving the nominal LP. In our Shortest-Path example, there are specialized algorithms that solve that LP problem EXTREMELY fast (e.g. a vectorized Dijkstra’s algorithm), but we do not know of any specialized algorithms for solving the QP to high accuracy. Generic solvers are orders of magnitude slower.
>
> #### ***Randomized Smoothing***
> We have already compared to a convexification of a randomized smoother (the Fenchel-Young Loss from [1]).  As seen in Fig. 2 and the new experiments (See Global Response Doc) our method can outperform this method empirically because it does not guarantee learning a best-in-class policy.
>
> On the other hand, the reviewer may have instead intended the “DPO” procedure from [1] (see also [29] in Section 3.4.3.), which replaces $\hat \pi(t)$ with a smoothed version
> $\mathbb E_\xi[\hat \pi(t + \sigma \xi)]$ and then attempts to solve the decision-loss with this smoothed version.  [1] provides no theoretical guarantees on performance. Moreover, one can check the resulting loss is still non-convex (take $\mathcal Z = [-1, 1]$, $Y = \{-1, 1\}$) . In THEORY, its gradients are Lipschitz (an advantage).  We say in theory, because in practice, the method uses monte carlo (see “Practical Implementation” on pg. 5 of [1]) with a small number of samples, and when replacing the expectation with a finite sum, the smoothness is lost.
>
> Thus, it is not clear that this optimization is better behaved than our proposal. Numerical experiments from [29] suggest that the performance is poor because of this monte carlo sampling. They write in Section 6 “Notably DPO was not shown because its overall subpar performance,” i.e., it was omitted from all plots.
>
> Finally, if the reviewer's concern is our lack of Lipschitz gradients, note that since we represent our loss as a difference-of-convex functions, we can “out-of-the-box” leverage [existing results on smoothing for DC functions](https://arxiv.org/abs/2104.01470). These results relate stationary points, and the global minimum of the smoothed function to the original function.
>
> **To summarize**
>  - NP-Hardness shows that ANY approach to this problem must either solve a hard optimization problem or cannot consistently learn best-in-class policies. We argue that although the problem is NP-Hard, our method finds high-quality, sub-optimal solutions.
>  - Regularization approaches [30] introduce significant computational challenges, are still non-convex, and do not induce Lipschitz gradients.
>  - We have already compared to the Fenchel Young Loss in the paper (a convexified randomized smoothing approach) and found this loss might not find a best-in-class policy (Fig 2 and Global Response Doc).
>  - Direct randomized-smoothing of the policy (DPO) induces a non-convex loss and does induce Lipschitz gradients (in theory).  However, prior work suggests that the challenges of Monte Carlo outweigh this benefit, and the method has poor performance.
>  - We can combine our method with existing approaches for smoothing DC function to recover Lipschitz gradients out of the box.
>
> ### ***Weakness 4: Additional experiments***
>  - Fig. 1 has a continuous region, synthetic data
>  - Shortest path experiments are combinatorial, synthetic data
>  - *NEW* portfolio experiment (continuous region), real data
>
> ### Questions:
> 1. Yes, the lemma already holds in this case (see our previous response on Danskin’s theorem).
> 2. Typo: Should say $Y$, not $Y_j$

---

> > ### Comment · Reviewer_oPp7 · 2024-08-12
> >
> > Thank you for your detailed response, which partially addresses my concerns. The new empirical results look promising, and I am hopeful of increasing the score.
> > As for Line 148 and eq 4, can you derive the directional gradient more carefully using Danskin's theorem, line by line? Also, please do not cite a blog post, please use more formal citations.

---

> ### Author Response · Authors · 2024-08-12
> **Proof for Eq 4**
>
> Thank you for your response! We are glad to hear about the positive feedback on our new empirical results. We apologize for the confusion about the blog post. The citation we would use in a galley proof is Prop B.22,
>
> *Bertsekas DP. Nonlinear Programming. 2nd ed. Athena Scientific; 1999.*
>
> We provided the website for the rebuttal period as it provides the same version of Danskin's theorem as the above textbook, but the textbook might not be available to everyone. In particular, the blog post copies the result nearly verbatim from the textbook, modulo some light formatting.  (The blogpost also cites the textbook above.)
>
> **Line by line proof of Eq. (4)**
>
> To more clearly match the notation of the reference, first rewrite
> $V(t) \equiv \min_{z \in \mathcal Z} t^{\top}z = - \max_{z \in \mathcal Z} -t ^\top z$, and define $\phi(t, z) \equiv -t^\top z$ and $f(t) \equiv \max_{z \in \mathcal Z} \phi(t, z)$.
> Then with these new notations, $V(t) = -f(t)$ and $\hat \pi(t) \in \arg \max_{z \in \mathcal Z} \phi(t, z)$.  (We provide the transformation because we are dealing with a "min" and the theorem is for a "max.")
>
> Note $\mathcal Z$ is a compact set, $\phi(t, z)$ is convex in its first argument, and $t \mapsto \phi(t, z) = -t^\top z$ is (everywhere) differentiable in $t$ for any $z$ because it's just a linear function.
>
> Let us first consider the case where there is a unique maximizer at $t_0$, i.e., $\hat \pi (t_0)$ is unique. Recall that $t \mapsto \phi(t, z)$ is differentiable in $t$ for all $z$, and in particular, is differentiable in $t$ at $z = \hat \pi(t_0)$. By part 3 of the blog post (equiv. the statement "If Z(x) consists of a unique point $\bar z \ldots$ " in the Bertsekas textbook),  we have that
> $
> \nabla f(t_0) = \frac{\partial \phi(t, \hat \pi(t_0) ) }{\partial t} = -\hat \pi(t_0),
> $
> from the definition of $\phi$.
>
> Then, since $V(t) = -f(t)$, we conclude that $\nabla V(t_0) = - \nabla f(t_0) = \hat \pi(t_0)$ for any $t_0$ where $\hat\pi(t_0)$ is the unique optimizer.  Hence the map $\lambda \mapsto V(t_0 + \lambda y)$ is differentiable (in $\lambda$), and by the chain-rule, $\frac{\partial}{\partial \lambda} V(t_0 + \lambda y) = \langle \nabla V(t_0 + \lambda y), \frac{\partial}{\partial \lambda } (t_0 + \lambda y) \rangle =  y^\top \nabla V(t_0 + \lambda y) = y^\top \hat \pi(t_0 + \lambda y)$.  Evaluating at $\lambda = 0$ proves Eq. 4 in this case.
>
>
> We now prove the statement when $\hat \pi (t_0)$ is not the unique maximizer.  Recall again that $t \mapsto \phi(t, z) = -t^\top z$ is differentiable in $t$ for all $z$, and the derivative $\frac{\partial \phi}{\partial t} = -z$ is continuous in $z$ for all $t$ (because it's just a linear function).
>
> Hence, by part 4 of the blogpost (equiv. part b of the Bertsekas Textbook) the set of subgradients of $f(t_0)$ is
>
> $\partial f(t_0)
> \ = \ \text{conv} \\{ \frac{\partial \phi(t, z)}{\partial t} \mid_{t = t_0} : z \text{ is a solution to }  \max_{z \in \mathcal Z} \phi(t_0, z) \\}
> \ = \
> \text{conv} \\{ -z : z \text{ is a solution to }  \max_{z \in \mathcal Z} \phi(t_0, z) \\} .$
>
> Thus, since $\hat \pi (t_0) \in \arg\max_{z \in \mathcal Z} \phi(t_0,z)$, we have $-\hat \pi (t_0)$ is a subgradient of $f(t_0)$.  This implies that for any $\lambda$,
> $
> f(t_0 + \lambda y) - f(t_0) \geq -\lambda \hat \pi(t_0)^\top y$.
> Recalling $V(t) = -f(t)$ and multiplying by $-1$ shows,
> $
> V(t_0 + \lambda y) - V(t_0) \leq \lambda \hat \pi(t_0)^\top y$, i.e.,
> $\hat \pi(t_0)^\top y$ is a subdifferential of $\lambda \mapsto V(t_0 + \lambda y)$. This concludes the proof of Eq. 4.

---

> > ### Comment · Reviewer_oPp7 · 2024-08-13
> >
> > Hi, thank you for the detailed reply.
> >
> > When V is nonsmooth, you showed $\hat{\pi}(t_0)^\top y$ is a subgradient, and the finite-difference (line 152,153) is approximating this value. But which $\hat{\pi}(t_0)$ is specifically chosen? As h converges to zero, the finite-difference must converge to a specific limit, and hence there must be a specific  $\hat{\pi}(t_0)$. Should you consider the directional derivative where $\hat{\pi}(t_0)$ is the one that aligns with $y$ most?

---

> ### Author Response · Authors · 2024-08-13
>
> Thank you for the positive signals and the new question.  We apologize for editing this response; we discussed internally and think we now better understand the heart of your question.  (If we still misunderstood, please accept our apologies. We are eager to  clarify once we better understand the question.)
>
> How we currently understand your question is:  Fix a $t_0$ such that $\hat \pi(t_0)$ is **not** the unique optimizer. Then, we have shown that $y^\top\hat \pi(t_0)$ is **a** subgradient of $\lambda  \mapsto V(t_0 + \lambda y)$ at $\lambda = 0$.  It's also clear that the finite difference $\frac{1}{h} (V(t_0) - V(t_0- hy))$ approximates **a** subgradient of this function at $\lambda = 0$.  Why is it that these are the same two subgradients (since there are multiple subgradients)?  In other words, why is it that $\lim_{h\rightarrow 0} \frac{1}{h} (V(t_0) - V(t_0 - hy)) = \hat \pi(t_0)^\top y$?
>
> This is an excellent and subtle question that highlights the role of Assumption 3.1 in our results.
>
> First, it is **not** the case that we can guarantee that $\lim_{h\rightarrow 0} \frac{1}{h} (V(t_0) - V(t_0 - hy)) = \hat \pi(t_0)^\top y$.  This is a ``path-by-path" requirement that is very strong.
>
> Why is this not a problem for our results?  Note, Eq. (4) is meant to be motivation (it does not occur as a formal theorem or in a proof).  It illustrates the intuition behind our PG losses.  The formal result is given in Lemma 3.2 (and subsequent results that build on it). The key idea is that although the above limit doesn't hold path by path, under Assumption 3.1, it **does** hold in expectation, i.e.,
> $\lim_{h\rightarrow 0} \mathbb E[ \frac{1}{h}(V(f(X)) - V(f(X) - hY))] = \mathbb E[ \ell(f(X), Y)]$.  (Lemma 3.2 actually proves a stronger statement by explicitly giving the rate.) Here the role of $t_0$ is played by $f(X)$ which is random. Holding in expectation is a weaker requirement, and since the ERM approximation concentrates at its true expectation uniformly (Thm 3.4 and Thm 3.7), it's enough that it holds in expectation.
>
> This of course raises an interesting question of whether one could make a stronger assumption than Assumption 3.1 and derive a path-by-path result.  We have not explored this idea.
>
> Did we correctly understand your question?  We're happy to clarify further however we can.

---

> > ### Comment · Reviewer_oPp7 · 2024-08-14
> >
> > My main point is you can use the definition of the directional derivative of a convex function to find out the specific $\hat{\pi}(t_0)$, then the whole thing can go through. I don't understand why this is difficult.
> >
> > Although I am still a bit skeptical about the argument and proof of Lemma 3.2, the conclusion of Lemma 3.2 appears reasonable, as the nonsmooth point has a zero measure and when you do the integration, you should expect a smooth behavior.

---

> > > ### Comment · Reviewer_oPp7 · 2024-08-14
> > >
> > > Overall, I believe the paper's revision shows some improvement, and I am inclined to raise the score based on the updates. However, I still believe the authors need to put significant effort into making their paper more rigorous.

---

### Official Review · Reviewer_LgZs · 2024-07-12

**Soundness:** 3
**Presentation:** 3
**Contribution:** 3
**Rating:** 5
**Confidence:** 3

**Summary:**

This paper considers a predict-then-optimize framework for solving contextual optimization problems, in particular for the case where the set of decisions is combinatorial or polyhedral, or when the loss is non-differentiable. They define a family of surrogate losses that connect the loss to the directional derivative of a plug-in function, and use zero-th order gradients to approximate the derivative. Simple numerical examples are provided.

**Strengths:**

The paper is well written and considers a principled approach.

**Weaknesses:**

The role of $h$ in the approximations of $\ell$ appearing in Theorem 2.1 should be made clear -- $\hat\ell^b$ and $\hat\ell^c$ have not been defined. Please give details of the proof of part c).

The main weakness is that the method appears to work well on a very simple synthetic problem. The performance appears to deteriorate on a slightly more complex problem, and then no interesting example is provided. I feel that for this outlet, this is a major weakness and would like to see how the method performs on a real world and/or large-scale problem. There are a number of such examples in the cited supporting literature.

**Questions:**

Can you please address the issues above?

**Limitations:**

Limitations are not addressed. Potential negative societal impact is negligible.

---

> ### Author Rebuttal · Authors · 2024-08-06
>
> ### **Clarifying Minor Weaknesses**
> We believe the reviewer meant Lemma 2.1, as there is no Theorem 2.1.
>
> #### **Role of $h$**
> The role of h is intuitively described on top of pg. 5 Line 154 (i.e. before Lemma 2.1).  This is further elaborated (quantitatively) after Corollary 3.3 (pg. 6 Line 214).
>
> #### **Definitions of Key Surrogates**
> The quantities $\hat \ell^b$  and $\hat \ell^c$ are defined on the bottom of pg. 4 (line 152), which occurs just before Lemma 2.1.
>
> #### **Proof of part c**
> We are happy to add the details of part c) in a gallery version. Here are the details for you to verify:  (As an aside, there are many versions of Danskin’s Theorem under different regularity conditions.  [This version](https://statisticaloddsandends.wordpress.com/2022/11/10/what-is-danskins-theorem/) is sufficient for the proof below.
>
> From the definition of $\hat \ell^b(t, y)$,  we have
> $\nabla \hat{\ell}^b(t, y) = \frac{1}{h} ( \nabla_t V(t) - \nabla V(t - hy) )$.
>
> We next evaluate each of the "gradients" on the right using Danskin’s Theorem. We say "gradients" because, as we will show, these are gradients when $\hat \pi(t)$ and $\hat \pi(t -hy)$ are unique, and are subgradients otherwise.  First, we validate the conditions of Danskin's Theorem for the first gradient.  Specifically,
> $V(t) = \max_{z \in \mathcal Z} \langle t, z \rangle$.
> The function $\phi(t, z) = \langle t, z \rangle$ is continuous and differentiable by inspection, and $\mathcal Z$ is compact by assumption.
> Thus, the conditions of Danskin’s theorem are met, and
> $\nabla_t V(t) = \hat\pi(t)$, where the left side is a gradient if $\hat \pi(t)$ is unique, and a subgradient otherwise (see part iv of above mentioned reference).
>
> We can treat $\nabla_t V(t - hy)$ in a similar fashion. Combining proves part c) for $\hat \ell^b$.  The proof for $\hat \ell^c$ is similar.
>
> ### **Re Weaknesses: Experimental Evaluation**
> Thank you for pushing us in this direction. In the Global Response Document, we’ve added two additional experiments: i) a harder instance of a shortest-path problem and ii) a portfolio optimization problem with **real data** and a low signal-to-noise ratio. In both cases, our method has an advantage over all baseline methods.
>
> #### ***New, Harder Shortest Path Instance***
>
> The difference between this new shortest path and the original shortest path instance is the data generation. In the original shortest path instance (following [6] and others), the costs of the arcs are exchangeable with respect to the network. There’s no special relationship between these costs and their location in the network. (For example, the edges of the square aren't systematically more expensive than internal roads.)  Consequently, many candidate paths have similar costs, and the problem is arguably not too difficult. That is why many baselines perform similarly.
>
> In our new instance, we generate the arc costs in a way that depends on the network. Specifically, we first embed two “good” paths along the diagonal (a safe one and a risky one) (see Global Response Doc), then ensure that any other path has a high cost, and then add noise to try and hide the “good” paths and confuse the safe and risky one. This is a more challenging setting because each method must first identify the good paths, and then choose between them to do well. As you can see, our approach has an edge in performance for large enough n.
>
> #### ***Portfolio Optimization***
> We study the same problem as [6, 26, 32] but use **real data**, specifically the 12 Fama French Industry Sector indices from the [Fama French Library](https://mba.tuck.dartmouth.edu/pages/faculty/ken.french/data_library.html). These indices represent returns of different asset classes and realistically mirror the asset allocation problem faced by wealth managers. We sample a month $t$ at random from the last 10 years, and let $Y = r_t$ be the return of the $d=12$  indices, and let $X = r_{t-1} + \mathcal N(0, 0.5 \Sigma)$  ($p=12$) where $\Sigma$ is the covariance of $r_t$.  The additional noise lowers the signal-to-noise ratio while maintaining the correlation matrix of $X$ and makes the problem harder.  See Global Response document.
>
> Because of limited computational resources, we only present the strongest benchmarks (SPO+, FYL, 2Stage PtO, and our method).  We again see that we have a distinct advantage.
>
> We're happy to include these additional experiments in a galley version to strengthen the empirical evaluation of the methods.
>
> #### ***Aside: On Value of Empirical Evaluation***
>
> Finally, we stress that the benchmarks above have NO theoretical guarantees in misspecified settings. We believe offering a theoretically justified surrogate for misspecified settings is interesting in its own right, beyond its empirical evaluation.
>
>
> ### Limitations
> Limitations are discussed on pg. 3 Line 76. We’re happy to label this discussion more clearly with a section header “Limitations” in a gallery proof it would help it to stand out.

---

> > ### Comment · Reviewer_LgZs · 2024-08-11
> > **Response**
> >
> > I have read the response, and it has helped clarify things. I am satisfied that these additions have improved the paper, and adjusted my score accordingly

---

### Official Review · Reviewer_CTUs · 2024-07-14

**Soundness:** 2
**Presentation:** 2
**Contribution:** 2
**Rating:** 5
**Confidence:** 3

**Summary:**

This paper addresses the predict-then-optimize problem by proposing a new family of surrogate loss functions. The key motivation is derived from Danskin's Theorem, which connects the expected downstream decision loss with the directional derivative of a particular plug-in objective. This objective is then approximated using zero-order gradient methods. The paper includes numerical experiments conducted on both a synthetic environment and a shortest path problem.

**Strengths:**

1. The paper is well-motivated.

2. The properties of the proposed surrogate loss are thoroughly derived.

3. Theoretical analysis shows that the approximation error of the proposed loss diminishes as the number of samples increases. Consequently, it can outperform existing surrogate losses even in misspecified settings.

**Weaknesses:**

1. The experimental section of the paper is relatively weak.

2. While the proposed method performs well in a simple synthetic environment, it does not demonstrate a clear advantage over FYL and SPO+ in the shortest path problem.

3. More experiments are needed to illustrate the advantages of the proposed method in real-world scenarios.

**Questions:**

1. How should the parameter h be selected in practice? Have you empirically studied how it affects performance?

2. In the shortest path experiment, why does FYL perform so well? Theoretically, the PG losses should be superior to FYL. Could you provide insights into this discrepancy?

**Limitations:**

I would suggest the paper add a limitation section.

---

> ### Author Rebuttal · Authors · 2024-08-06
>
> ### **Re Weaknesses: Experimental Evaluation**
> Thank you for pushing us in this direction. In the Global Response Document, we’ve added two additional experiments: i) a harder instance of a shortest-path problem and ii) a portfolio optimization problem with **real data** and a low signal-to-noise ratio. In both cases, our method has an advantage over all baseline methods.
>
> #### ***New, Harder Shortest Path Instance***
>
> The difference between this new shortest path and the original shortest path instance is the data generation. In the original shortest path instance (following [6] and others), the costs of the arcs are exchangeable with respect to the network. There’s no special relationship between these costs and their location in the network. (For example, the edges of the square aren't systematically more expensive than internal roads.)  Consequently, many candidate paths have similar costs, and the problem is arguably not too difficult. That is why many baselines perform similarly.
>
> In our new instance, we generate the arc costs in a way that depends on the network. Specifically, we first embed two “good” paths along the diagonal (a safe one and a risky one) (see Global Response Doc), then ensure that any other path has a high cost, and then add noise to try and hide the “good” paths and confuse the safe and risky one. This is a more challenging setting because each method must first identify the good paths, and then choose between them to do well. As you can see, our approach has an edge in performance for large enough n.
>
> #### ***Portfolio Optimization***
> We study the same problem as [6, 26, 32] but use **real data**, specifically the 12 Fama French Industry Sector indices from the [Fama French Library](https://mba.tuck.dartmouth.edu/pages/faculty/ken.french/data_library.html). These indices represent returns of different asset classes and realistically mirror the asset allocation problem faced by wealth managers. We sample a month $t$ at random from the last 10 years, let $Y = r_t$ be the return of the $d=12$  indices, and let $X=r_{t-1} + \mathcal N(0, 0.5 \Sigma)$ be the previous month return plus Gaussian noise ($p=12$). Here $\Sigma$ is the covariance of $X$.  The additional noise lowers the signal-to-noise ratio while maintaining the correlation matrix of $X$.   See Global Response document.
>
> Because of limited computational resources, we only present the strongest benchmarks (SPO+, FYL, 2Stage PtO, and our method).  We again see that we have a distinct advantage.
>
> We're happy to include these additional experiments in a galley version to strengthen the empirical evaluation of the methods.
>
> #### ***Aside: On Value of Empirical Evaluation***
>
> Finally, even if the reviewer feels our method performs comparably to existing benchmarks, it should be stressed that those benchmarks have NO theoretical guarantees in misspecified settings. We believe offering a theoretically justified surrogate for this setting (even with comparable performance) is interesting in its own right.
>
> ### **Question: How to select $h$?**
> We selected $h$ using hold-out validation set of size $200$ in our experiments.  In general, we found the method insensitive to choice of $h$ as long as it was reasonably small. Please see plot on Global Response Doc.
>
> ### **Question: Why does FYL perform well in (original) Shortest Path?**
> This is a great question.  Our current conjecture is that it is a combination of two features:
> 1. Since arc costs are exchangeable across the network, there are many good candidate paths, and FYL is finding one of them.
> 2. By looking at its gradient, we argue that FYL essentially searches for a policy $T(\cdot)$ such that $\hat \pi(T(X)) \approx \hat \pi ( Y) $.  Notice, this isn't the same as the oracle optimality condition, which would seek a policy such that $\hat \pi(T(X)) \approx \hat \pi (f^*(X))$.
>
> These two conjectures informed our new (harder) shortest-path example where we i) embed only two good paths (risky and safe,) so that to perform well, a method must identify these two paths among all other paths and choose between them and ii) Lower the signal-to-noise ratio so that $Y$ is further from $f^*(X)$, and, hopefully, $\hat\pi(Y)$ is more distinct from $\hat\pi(f^*(X))$.  As seen in Global Response doc, this does seem to affect FYL's performance.
>
> We also note that FYL performs surprisingly poorly in our portfolio allocation experiment. (See Global Response doc)
>
>
> ### **Limitations**
> We discuss limitations on pg. 3 Line 76. We’re happy to label this discussion more clearly as with a section heading saying “Limitations” in a gallery proof if it would help it to stand out.

---

### Official Review · Reviewer_TiWp · 2024-07-17

**Soundness:** 4
**Presentation:** 3
**Contribution:** 3
**Rating:** 7
**Confidence:** 4

**Summary:**

This paper proposes a family of "perturbation gradient" losses for Predict-than-Optimize (PtO) that, if optimized for, can lead to best-in-class performance, even under model misspecification. On the theoretical side, this paper provides risk bounds that build on past theoretical work in PtO + the literature on using perturbation-based approaches for estimating out-of-sample performance. Importantly, it shows that the excess risk goes to zero when the number of data points $n \to \infty$. On the empirical side, they show that their loss functions outperform others from the literature in one synthetic domain and perform comparably to other loss functions in one domain from a popular benchmark.

**Strengths:**

* The paper identifies and addresses an important problem in the PtO literature--performance guarantees under model misspecification.
* I'm not a theoretician, but the theoretical results seem non-trivial and relevant to practice.
* On the empirical side, they compare to a reasonable set of baselines from the literature.

**Weaknesses:**

I have reviewed this paper in the past, and my two major issues were (a) that it overclaimed and (b) it had weak empirical analysis. While the paper has improved significantly on both fronts, I still have some gripes:
* _Regarding Claims:_ While the paper has added a paragraph about the approach's limitations in the introduction, I'm not sure that I understand it. My issue with the approach is that if the _true_ loss is non-convex in the policy parameters, it will be hard to optimize for, even in the limit of infinite data when $n \to \infty$. This isn't the same as the issue you describe for small $n$ in Figure 2(b), or even the statistical complexity issue of cleverly choosing $h$ as discussed in Section 3.2. It's that if the true loss $\ell(t(\theta), y)$ is piecewise constant, then for small $h$ the PG-loss $\hat{\ell}^b_h$ will indeed be close to $\ell$ but that means it will be close to piecewise constant and, as a result, hard to optimize for. The current theory assumes that you can optimize for $\hat{\ell}^b_h$ but not $\ell$, but that's a big assumption that should be discussed. Using the analogy of the ramp loss from the introduction, if your initial prediction is sufficiently far from $t = 0$, the gradient of the ramp loss will still be 0 (same as the $sgn$), and you won't be able to use first-order methods to learn a good model $\hat{f}$. Perhaps this is something that comes under your buckets of (a) the difficulty of optimizing for a "difference of convex functions" in the introduction or (b) the bias-complexity tradeoff in choosing $h$ that you allude to in the conclusions, but I couldn't immediately see the connection. Could you talk more about this?
* _Empirical Evaluation:_ I appreciate that you have included $SPO+$ and $PFYL$ as baselines, and also added a PyEPO domain. However, the results aren't conclusive even under significant model misspecification (e.g., you do no better than PFYL in either case, even when PFYL has no guarantees under misspecification, and SPO+ does roughly the same for uniform noise). Given that you've implemented one domain, running tests on the other domains in the benchmark should be fairly easy. Have you run those experiments? What do they look like?

**Questions:**

Could you address my comments in the weaknesses section?

**Limitations:**

The paper does an okay job of addressing the limitations of the proposed approach.

---

> ### Author Rebuttal · Authors · 2024-08-05
>
> ### **Overview**
> We’d like to recall it is NP-Hard to optimize the (true) decision-loss over linear functions [6], essentially because it generalizes binary classification. Hence, any method (including ours) that aims to learn a best-in-class policy for all data generation mechanisms MUST also be NP-Hard.  That said, theory aside, not all NP-Hard problems are created equal. Some, e.g., knapsack or bin-packing, admit “practically efficient” algorithms that solve most real-world instances in reasonable amounts of time. In contrast, others (e.g., non-metric TSP) are so hard we do not even have reliably good heuristics for large-scale instances. What we are trying to argue in our paper is that although optimizing our surrogate is NP-Hard (as it MUST be), even simple gradient descent algorithms recover very high-quality local minima that are suitable for applications.
>
> ### **Specific Questions Re Claims**
> You are correct that part of the difficulty is that far away from the “heart” of the function, the true loss is flat, and, hence, our loss is flat. To be more concrete, let’s focus on the ramp loss. Basically, for a single data point $(X_i, Y_i)$, if $T(X_i)$ is more than $O(h)$ from zero, both losses will be flat. This challenge is shared by other losses, e.g., the losses proposed in [30, 24].
>
> As you’ve also observed, this issue connects both to a) the difference-of-convex function representation and b) the bias-complexity tradeoff of choosing h.
>
> #### _The Bias-Complexity Tradeoff_
> As above, in the case of the ramp loss, if we have a single data point $(X_i,  Y_i)$ and if $T(X_i)$ is more than $O(h)$ from zero, the loss is flat. However, when we have $n$ data points, the empirical loss is only flat if we are more than $O(h)$ away from ALL data points. Hence, for large $n$, our loss is unlikely to be flat in regions of high data density. Moreover, the larger we make $h$, the less likely we’ll end up in a bad region during gradient descent (provided we initialize at a “smart” point, see below). One of the takeaways of our theoretical analysis is characterizing precise conditions and how large we can make $h$ (to minimize the chances of reaching a flat region) while still guaranteeing a good enough approximation to learn the best-in-class policy, and how this should scale with $n$.
>
> #### _Difference of Convex Function Representation_
> We represent our loss explicitly as a difference of convex functions. This means we know A LOT about the structure of the loss landscape.  DC optimization is a growing field [F1], [F2],  and there are recent works on how to smooth DC functions [F3], [F4], [F5] to improve computational performance, and even new algorithms for identifying smart starting points for multistart gradient ascent [F6].  (These could serve as the aforementioned “smart” starting points to ensure your algorithm doesn’t get stuck in the flat parts far away from the “heart” of the function.)  By contrast, the loss landscape of the original decision loss is much less understood, and so it’s less obvious how to optimize it directly (even heuristically).
>
> #### **Summary**
> We are not trying to mislead; we ARE replacing one NP-Hard optimization problem with a different NP-Hard optimization problem. However, in practice, some NP-Hard problems admit algorithms that find high-quality solutions very efficiently for practical instances, and we argue our loss leads to one such problem. Trying to quantify this “improved tractability” is necessarily subtle.
>
> #### _Addendum on Non-Convexity_
> Finally, we’d politely point out that the issue of convexity vs. non-convexity is often moot in applications. When using a nonlinear hypothesis class (e.g., a neural network with more than 1 layer), even surrogates like SPO+ and PYFL induce non-convex loss functions. These more powerful hypothesis classes are often preferred in practice, and, for these settings, optimizing these losses is theoretically no easier than optimizing our surrogate.
>
> ### **Specific Questions Regarding Empirical Evaluation**
> Thank you for pushing us.  Based on feedback from you and the review team, we have added two experiments: 1) A Harder instance of Shortest Path where we've hidden good paths in the network and 2) A Portfolio Optimization Example with low signal to noise ratio.  In both of these "harder" settings, simple gradient descent procedures (ADAM) on our loss recovers local minima that substantively outperform benchmarks.  ***See Global Rebuttal Document.***  Obviously, leveraging tools from the DC literature, one might be able to further improve upon these solutions.
>
> Finally, in addition to the observed empirical benefits of our methods, we stress that existing
> benchmarks have NO theoretical guarantees in misspecified settings. Offering a theoretically justified surrogate for misspecified settings is interesting in its own right.
>
>
>  - F1: https://link.springer.com/article/10.1007/s11081-015-9294-x
>  - F2: https://link.springer.com/article/10.1007/s10107-018-1235-y
>  - F3: https://arxiv.org/abs/2104.01470
>  - F4: https://ieeexplore.ieee.org/document/9304514
>  - F5: https://www.sciencedirect.com/science/article/pii/0022247X91901875
>  - F6: https://pubsonline.informs.org/doi/abs/10.1287/ijoc.2022.1238

---

> ### Comment · Reviewer_TiWp · 2024-08-13
> **Response to Rebuttal**
>
> Thank you for clarifying the theoretical contributions and adding new experiments. **While I still have a few questions, I will increase my score to a 7 and recommend acceptance.** I think that (even without the theoretical properties) the paper has proposed a novel predict-then-optimize surrogate and shown improved performance on (variants) of standard domains in the literature. This is the bar to which papers in this domain have been held in the past.
>
> As for my remaining questions/concerns:
>
> ### Experiments
>
> **[Q1]**  Why didn't you run experiments on the other domains from PyEPO (knapsack and TSP) or even existing implementations of the Portfolio Optimization problem? I find this a bit confusing because it should have been easier than creating the new domains that you have presented in your paper. Additionally, it would allow us to compare the effectiveness of the proposed approach to a much larger set of surrogates, which have also been evaluated on these more standard datasets. Even if it seems like PGC doesn't really beat SPO+ or alternatives, it would be (IMO) useful to know.
>
> **[Q2]** How are the features generated in the new shortest path example? In the old version of the problem, the features for each edge seem to be generated independently (from a normal distribution). But if this were the case, I don't see how any model would be able to isolate the "safe" paths based on just the features.
>
> Also, I have run experiments on the portfolio optimization domain from [26, 32] and have never seen improvements as large as those you've found in your paper. I hope you release your code, and look forward to investigating this version of the problem in more detail!
>
> ### Theoretical Properties
>
> I think these clarifications are super useful, and I hope that they will be included in the final version of the paper. However, I still have some basic questions.
>
> **[Q3]** When you say:
>
> > "What we are trying to argue in our paper is that although optimizing our surrogate is NP-Hard (as it MUST be), even simple gradient descent algorithms recover very high-quality local minima that are suitable for applications."
>
> > "However, in practice, some NP-Hard problems admit algorithms that find high-quality solutions very efficiently for practical instances, and we argue our loss leads to one such problem."
>
> I don't understand how your theorems show this (although your experiments do). From my understanding, your theorems show that _if you can optimize the surrogate loss_ for some value/schedule of $h$, you will be able to optimize the true loss. However, it says nothing about being able to optimize for the surrogate with gradient descent, which seems to be critical to these arguments that you're making above. Am I misunderstanding something? Could you also link me to the theorem statement that shows:
>
> > One of the takeaways of our theoretical analysis is characterizing precise conditions and how large we can make $h$ (to minimize the chances of reaching a flat region) while still guaranteeing a good enough approximation to learn the best-in-class policy, and how this should scale with $n$.
>
> **[Q4]** When you say:
>
> > However, when we have $n$ data points, the empirical loss is only flat if we are more than away from ALL $n$ data points. Hence, for large $n$, our loss is unlikely to be flat in regions of high data density.
>
> This still does not guarantee that you will be able to optimize for the surrogate loss function. What stops gradient descent from reaching a local optimum in which you do better for the subset of points that have a non-zero gradient and do badly for those with zero gradients?
>
> **[Q5]** Also, when you talk about optimizing for the PG losses as solving a "difference in convex functions" problem, (based on my skimming the abstracts) the papers that you link seem to use some sort of clever smoothing to solve the problem. However, you don't seem to be smoothing your PG losses. Why can you still solve the optimization problem for PG losses, then?

---

> ### Author Response · Authors · 2024-08-14
>
> Thank you for the positive feedback.  To be clear (for future AC's that might be skimming):  when you say "even without the theoretical properties," did you mean
> 1) That the empirical/methodological contributions merit publication on their own, and the theoretical contributions are "bonus" or
> 2) that you have some unresolved questions about the proof/statements of the theoretical results?
>
>
> If it's 2), please flag the questions for us and we are happy to address them. From our own viewpoint, we provide some of the first theoretical guarantees for best-in-class behavior in a misspecified setting using a surrogate that supports gradient descent, and we see this as an important contribution.
>
> **[Q1]**
> >Why didn't you run experiments on the other domains from PyEPO (knapsack and TSP) or even existing implementations of the Portfolio Optimization problem? ....
>
> We agree that more benchmarking is undoubteldy helpful.  Our choice of experiments in the response document was determined by 1) space constraints 2) requests from Reviewers LgZs and CTUs for real data and 3) Computational time limits in the rebuttal period.  We intend to present a full set of benchmarks in a journal version of the paper for researchers to use.
>
> More specifically, the PyEPO experimental set up for both knapsack and TSP is based on synthetic, random data. Given the other reviwer requests for real data, we thought this would add little to our (existing) synthetic data experiments.
>
> We *do* use a standard formulation of the portfolio optimization problem from [26].  We change the data set used to the Fama French dataset because we wanted a setting with high misspecification.  Indeed, for the QuandlWIKI dataset from [26, Table 1], 2 Stage MSE does almost as well as the best decision-focused methods, suggesting (to us) that the dataset is close to well-specified.  In other words, there's seemingly not a lot of "room" for *any* decision-focused method to shine.  We conjecture this might be because for daily stock returns i) the time scale is short enough that yesterday's stock price is a good predictor for today's stock price ii) the various stocks are very highly correlated. By contrast, the longer timescale of the monthly Fama-French returns make predictions more difficult and the different asset classes makes the signal weaker.
>
> **[Q2]**
> > How are the features generated in the new shortest path example?...
>
> In the original shortest path problem [6, 29], each problem instance is generated with 5 features $\mathbf{X} \in \mathbb{R}^5$ drawn from multivariate normal distribution and edge weight $i$ is $f_i(\mathbf{X}) = \frac{1}{3.5^6} \left( (\frac{1}{\sqrt{p}}\beta_i^{\top}\mathbf{X} + 3)^6 + 1 \right)$
> and $\beta_i \in \mathbb{R}^5$ are independently generated Bernoulli vectors.
>
> In the new experiment, we add a new feature so $\mathbf{X} \in \mathbb{R}^6$ and assume the new feature $X_6$ is drawn from a uniform distribution with support $[0,2]$. We modify $f_i$ for the two paths highlighted in the global response doc. For red path we let $f_i(\mathbf{X}) = 2$ for all $i$ on the path and for the blue path we let $f_i(\mathbf{X}) = 4X_6$ if $0 \le X_6 \le 0.55$ and $f_i(\mathbf{X}) = 2.2$ otherwise. Finally, for all other edges, we let $f_i(X) = \frac{1}{3.5^6} \left( (\frac{1}{\sqrt{p}} \sum_{j=1}^5 \beta_{ij}^{\top}X_j + 3)^6 + 1 \right) + 2.2$ which is the same as the original shortest path experiment but shifted up by $2.2$. This shift ensures the red and blue paths are better than the rest in expectation, and the best one depends on the value of $X_6$.  Finally, we add independent noise (Gaussian or Uniform) to all edges just as in original experiment.

---

> ### Author Response · Authors · 2024-08-14
>
> **[Q3]**
>
> > When you say:``What we are trying to argue $\ldots$"
> > Am I misunderstanding something?
>
> You are not misunderstanding. Our theory does say something ***slightly*** stronger -- namely because we prove uniform convergence, we've shown that if you can find a hypothesis $f(\cdot)$ that has low empirical PG loss, then it will also have low expected decision loss. So one need not perfectly optimize the PG Loss; just find a ``good enough" sub-optimal solution.  But again, our theory does not guarantee that gradient descent will necessarily find such a $f(\cdot)$.
>
> What we meant by our original comment (which you also correctly summarized) is that the ***empirical*** experiments suggest that simple gradient descent procedures do find high-quality, sub-optimal solutions.
>
> Unfortunately, in light of the NP-Hardness of the problem, it seems difficult (or impossible?) to formulate a theoretically rigorous tractability result that would apply generally. (This difficulty applies to ***any*** surrogate that achieves best-in-class performance, not just ours.)  So these empirical demonstrations are all we can (currently) offer. In many ways, this mirrors the state of the art with deep learning, where theory suggests the problem is hard/intractable, but empirical experience suggests we can reliably find high-quality local optima with (multi-start) stochastic gradient descent.
>
>
> **[Q3] Continued**
> > Could you also link me to the theorem statement that shows $\ldots$
>
> Happily!  Please see Theorems 3.4 and 3.7.
>
> Without the theorems, intuition suggests that if we want the empirical PG loss to well-approximate the expected decision loss as $n\rightarrow \infty$, we need $h_n \rightarrow 0$.  Indeed, any such sequence should suffice.  Since we want $h_n$ to be big (to avoid the flat regions), this suggests choosing a large $h$ that decays slowly. This is essentially the suggestion in [24], which advocates for very large $h$, like $h=10$. Experiments from [29] suggest this doesn't work well.
>
> By contrast, our Theorem 3.7 gives a tighter result and hence more insight.  It shows the error between the empirical PG loss and expected decision-loss is roughly $\tilde O(\min(h, 1/\sqrt n))$. Hence, taking $h$ larger than $O(1/\sqrt n)$ slows the convergence rate.  Thus, we might choose $h = 1/\sqrt n$, i.e., as large as possible without affecting the convergence rate. (See also line 242). Our theorems provide this kind of practical insight, and choosing $h$ in this manner drives (some of) our numerical improvements.
>
> A similar analysis holds for Theorem 3.4 and relates $h$ to the Rademacher complexity of the chosen class. See Line 227.

---

> > ### Author Response · Authors · 2024-08-14
> >
> > **[Q4]**
> > > When you say "However, when we have $\ldots$"  $\ldots$
> > > What stops gradient descent from reaching a local optimum $\ldots$ ?
> >
> > You are correct:  On its own, there is no guarantee we can find the global optimum of our surrogate by gradient descent, and the NP-Hardness result suggests we can't globally optimize any surrogate that consistently achieves best-in-class performance.  All we are hoping for is a high-quality local optimum.  This mirrors the case of training a neural network.
> >
> > One reason to *intuitively* believe that we should be able to find good local optima is that under Assumption 3.1, as $n\rightarrow \infty$, the decision-loss curves become smoother/more well-behaved. In other words, there are fewer "flat" locations and fewer bad local minima. See Right Panel of Fig. 2. In fact, implicit in the proof of Lemma 3.1, is the fact that (under Assumption 3.1) the function $t \mapsto \mathbb E[\ell(t, y)]$ (expected decision-loss) is differentiable with Lipschitz gradients. Since, by Theorem 3.4 and Theorem 3.7, the empirical PG loss looks closer and closer to this function, this gives us some hope for gradient descent methods for large $n$.  (Again, this is not a proof we can find the optimum, just intuition that we should converge to a stationary point.)
> >
> > We discuss this implicit fact on the bottom of pg. 5 with respect to how it affects approximation error, but we are happy to i) make this implicit fact explicit as a standalone lemma (with proof) and ii) connect this fact to the (intuitive) performance of first-order methods if you think it would help with the intuition.

---

> > > ### Author Response · Authors · 2024-08-14
> > >
> > > **[Q5]**
> > > > Also, when you talk about optimizing the PG Losses as solving a ``difference in convex functions" problem $\ldots$.  Why can you still solve the optimization problem or PG losses, then?
> > >
> > > We apologize for any confusion. First, in our experiments, we are not explicitly leveraging any special DC structure; we're just doing SGD. SGD can be run ``out of the box" without smoothing. Our comment indicated that exploiting the DC structure through specialized algorithms can only improve the empirical performance.
> > >
> > > Second, the most classical approach to DC problems is some form of DCA [F7], which solves a sequence of convex upper bounding problems to find a local optimum. DCA also does not require that the constituent DC functions be differentiable or smooth; it can also be applied ``out-of-the-box" in our case (but we didn't do this.)
> > >
> > > The references we gave on smoothing ([F3], [F4], [F5]) are recent works that argue that clever smoothing of the constituent functions does not affect stationary points/local optima. They are part of an area of research focusing on using first-order methods to optimize DC functions (see references).  They show that this smoothing induces various nice properties (Lipschitz differentiability, coercivity/level boundedness, etc.), and that, as a consequence, various first-order methods (on the smoothed function) converge to a stationary point of the original (non-smooth) function. Thus, we believe smoothing the PG loss in this way before running SGD is a promising area of future research. Again, intuition suggests that it can only improve our numerical performance, but we have not yet tried it.  Again, these results do not guarantee we find the global optimum, just that we should be able to identify a stationary point.
> > >
> > >  - [F7]: Lipp, Thomas, and Stephen Boyd. "Variations and extension of the convex-concave procedure." Optimization and Engineering 17 (2016): 263-287.

---

> > > > ### Comment · Reviewer_TiWp · 2024-08-14
> > > >
> > > > Thank you for the detailed feedback! As for your question about what I mean by "even without the theoretical properties," it's closer to 1 than 2. I am more of an empiricist, and while I appreciate the fact that these are some of the first theoretical results for this problem, I don't see how these results will impact practice; this is a complaint I also have with some deep learning theory, and the AC is free to overrule me. Specifically, the NP-Hardness of it all means that, unlike the (limited) regret bounds for other convex surrogates (e.g., SPO+), which you *can* easily optimize for, the fact that you aren't guaranteed to be able to optimize the PG-losses means that you can't give any real guarantees about the performance of your proposed surrogate. Empirically, you show that you actually achieve low PG-loss regret in practice and that it implies good predict-then-optimize performance, but you don't actually need any regret bounds to show that. This is why I feel like the theory is more of a "bonus"...
> > > >
> > > > As for the specific questions that I had:
> > > >
> > > > **[Q1]** That's fair! I look forward to your benchmarking results. Also, the "monthly" prediction being harder than the "daily" ones is an interesting insight and one that's quite testable on the QuandlWIKI dataset; I will try it out!
> > > >
> > > > **[Q2]** I see. So even though the features are "independently" generated, the features of edges that aren't on these two paths are just a "distraction". However, because the $X_6$ values for those edges are also drawn from $U[0, 2]$ (as opposed to, say, $X_6 = -1$), you will never be able to exactly pick the "safe path" (or even the "risky path"). But that's why you end up having such a large regret, even for PG losses, I guess.
> > > >
> > > > **[Q3]** Responded above. Let me know how you disagree with my opinion about the theoretical results.
> > > >
> > > > **[Q4]** Re: "losses getting smoother as $n \to \infty$", I broadly agree with your intuition. However, if we want to make things quantitative, I imagine that the number of samples grows exponentially with the number of feature dimensions for "a certain level of smoothness" because of the curse of dimensionality. In Fig: 2, things are nice and 1-d, but I don't know how well this would work for more complex problems...
> > > >
> > > > **[Q5]** Thank you for the additional information about the DC literature!

---

### Author Rebuttal · Authors · 2024-08-06

Attached is our global response document, which includes the following:

i) An updated shortest path experiment that embeds two "good" paths that methods must identify and choose between based on the context. This experiment increases the difficulty and reward of finding the oracle policy compared to the initial shortest path experiment. This allows us to show our PG Losses can learn better policies compared to existing surrogate benchmarks.

ii) We highlight how the choice of h affects the learned policy for the shortest path problem and see that for our choices it had minimum effect as long as h was sufficiently small.

iii) We introduce a new portfolio optimization experiment that was generated with **real** data. Our formulation follows existing benchmarks [6, 26, 32].  We again consider a linear objective of maximizing returns, but our feasible set is constructed with both a quadratic constraint and linear constraint. We plot the relative regret (lower is better) and show our approaches significantly out-perform existing benchmarks.

---

### Decision · Program_Chairs · 2024-09-25

**Decision:**

Accept (poster)

**Comment:**

This paper examines the prediction-then-optimize problem and introduces a novel Perturbation Gradient (PG) loss for decision-aware learning. The approach is innovative and shows promising performance compared to existing methods. The authors' rebuttals have successfully addressed most of the reviewers' questions.

For the camera-ready version, the authors should incorporate the following improvements:
- Incorporate the additional experimental results.
- Expand on the discussions regarding the complexity of optimizing the PG loss, as discussed with reviewers TiWp and oPp7.
- Discuss the choice of the $h$ parameter.
- Clarify the sub-differentiability issue raised by reviewer oPp7.
- Address the initialization for optimizing the PG loss, given its non-convex nature. The appendix notes that the SPO+ solution is used for initialization. This should be clarified in the experiment section, and the impact of different initializations (e.g., random initialization or initialization at the ETO or FY solutions) should be empirically demonstrated.